# Video-o3: Native Interleaved Clue Seeking for Long Video Multi-Hop Reasoning

Xiangyu Zeng [* 1 2]  Zhiqiu Zhang [* 1 2]  Yuhan Zhu [* 1 2]  Xinhao Li [* 1]  Zikang Wang [* 3 2]
Changlian Ma [1 2]  Qingyu Zhang [1]  Zizheng Huang [1]  Kun Ouyang [4]  Tianxiang Jiang [5 2]
Ziang Yan [6 2]  Yi Wang [2]  Hongjie Zhang [2]  Yali Wang [7 2]  Limin Wang [1 2]

https://mcg-nju.github.io/Video-o3

## Abstract

Existing multimodal large language models for long-video understanding predominantly rely on uniform sampling and single-turn inference, limiting their ability to identify sparse yet critical evidence amid extensive redundancy. We introduce **Video-o3**, a novel framework that supports iterative discovery of salient visual clues, fine-grained inspection of key segments, and adaptive termination once sufficient evidence is acquired. Technically, we address two core challenges in interleaved tool invocation. First, to mitigate attention dispersion induced by the heterogeneity of reasoning and tool-calling, we propose **Task-Decoupled Attention Masking**, which isolates per-step concentration while preserving shared global context. Second, to control context length growth in multi-turn interactions, we introduce a **Verifiable Trajectory-Guided Reward** that balances exploration coverage with reasoning efficiency. To support training at scale, we further develop a data synthesis pipeline and construct **Seeker-173K**, comprising 173K high-quality tool-interaction trajectories for effective supervised and reinforcement learning. Extensive experiments show that Video-o3 substantially outperforms state-of-the-art methods, achieving 72.1% accuracy on MLVU and 46.5% on Video-Holmes. These results demonstrate Video-o3's strong multi-hop evidence-seeking and reasoning capabilities, and validate the effectiveness of native tool invocation in long-video scenarios.

---

[*]Equal contribution  [1]Nanjing University [2]Shanghai AI Laboratory [3]Shanghai Jiao Tong University [4]Peking University [5]University of Science and Technology of China [6]Zhejiang University [7]SIAT, Chinese Academy of Sciences. Correspondence to: Limin Wang <lmwang@nju.edu.cn>.

*Proceedings of the $43^{rd}$ International Conference on Machine Learning*, Seoul, South Korea. PMLR 306, 2026. Copyright 2026 by the author(s).

## 1. Introduction

Multimodal Large Language Models (MLLMs) have achieved remarkable progress in recent years (Bai et al., 2025b; Li et al., 2026; Wang et al., 2025d; Feng et al., 2025). Although current models demonstrate strong performance on short video clips, extending these capabilities to long-form video understanding remains challenging. Long videos are characterized by abundant visual cues and intricate temporal dependencies, requiring models not only to precisely localize query-relevant moments but also to reason over these moments for accurate, query-specific understanding (Yu et al., 2025a; Zeng et al., 2025a). However, most existing approaches rely on uniform frame sampling followed by single-turn inference (Li et al., 2026; 2025b). Such strategies often dilute critical visual evidence within redundant background content, leading to both substantial computational overhead and degraded reasoning accuracy. In contrast, humans typically approach long videos in a goal-driven and exploratory manner: we first conduct a coarse-grained scan of the entire video and then iteratively focus on a small number of informative segments where the answer is most likely to be found. This observation naturally leads to a fundamental question: *can we endow MLLMs with a human-like **exploratory clue-seeking** capability to enable more efficient and accurate long video understanding?*

Although early "clue seeking + answer reasoning" prototypes have appeared (Yan et al., 2025a; Zhang et al., 2025b; Fu et al., 2025b), they rely heavily on hand-crafted heuristics and lack end-to-end optimization. Most methods decouple clue seeking from reasoning, training them as isolated single-turn modules without multi-step context sharing. At inference time, fixed search parameters further constrain the trajectory, reducing the process to simplistic single-turn localization or iterative single-clue refinement. As a result, these approaches fail to scale to long, complex videos that demand joint reasoning over multiple interdependent clues.

Compared to the aforementioned approaches, end-to-end training with native multi-turn tool invocation offers greater flexibility but also introduces substantially greater chal-

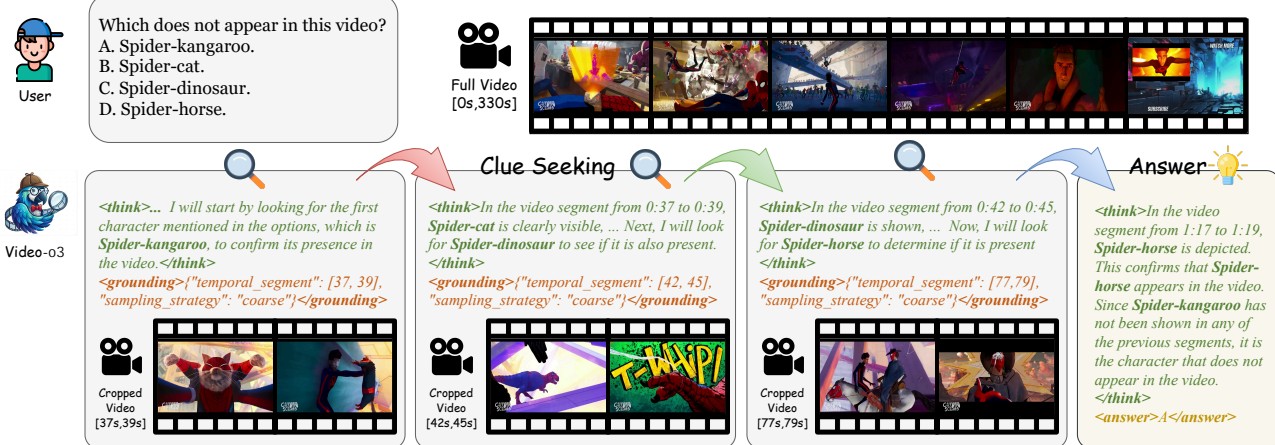

*Figure 1.* **Overview of our proposed Video-o3.** Guided by the query and current visual observations, Video-o3 actively identifies and localizes critical visual clues. It utilizes native interleaved tool invocation to capture video clips with dynamic quota. Following a detailed scrutiny of these local segments, the model autonomously decides whether to continue the search for further evidence or to conclude the reasoning process with a direct answer.

lenges (Lai et al., 2025; Zhang et al., 2025b). Unlike pipelined or modular designs, this paradigm must simultaneously support clue acquisition and multi-clue joint reasoning within a shared attention context, while autonomously deciding both the number of retrieval turns and the sufficiency of accumulated evidence. These requirements give rise to two fundamental challenges. *First, attention dispersion induced by heterogeneous tasks.* Native multi-turn reasoning operates over a shared context in which evidence seeking and reasoning are tightly interleaved. The entanglement of these heterogeneous processes can interfere with attention allocation, preventing the model from consistently focusing on the most informative tokens. *Second, contextual efficiency under strict budget constraints.* Each tool invocation incurs a significant token cost, making efficient context utilization critical. The model must therefore accurately localize relevant clues to avoid introducing irrelevant content, while also exercising reliable termination judgment to halt retrieval once sufficient evidence has been obtained, thereby preventing unnecessary and costly interactions.

To address these challenges, we introduce **Video-o3**, a novel framework that pioneers native interleaved tool invocation for long-video multi-hop reasoning. Video-o3 explicitly disentangles complex inter-clue dependencies and generates intermediate reasoning signals to guide structured exploration over extended temporal horizons. By dynamically invoking the VideoCrop tool, the model adaptively inspects fine-grained details within targeted segments. The final prediction is obtained through iterative revisitation of salient clue moments and principled aggregation of evidence distributed across disparate temporal intervals.

Specifically, we make the following three contributions: *First, to mitigate attention dispersion induced by heterogeneous tasks, we introduce Task-Decoupled Attention Mask-*

*ing.* This mechanism imposes explicit decoupling constraints on attention patterns for clue discovery and answer reasoning tasks, enabling an end-to-end model to concentrate on each stage of exploration while maintaining a shared global context. *Second, to improve contextual efficiency in multi-turn reasoning, we design a Verifiable Trajectory-Guided Reward.* We augment the answer reward with a trajectory-dependent multiplier that assigns higher reward to solutions achieving correct answers through more accurate clue localization and less redundant exploration paths. This formulation encourages the model to balance exhaustive exploration with computational efficiency, learning to terminate precisely when sufficient evidence has been accumulated. *Third, to overcome the bottleneck of data scarcity, we propose a scalable automated data synthesis pipeline.* Leveraging this pipeline, we construct Seeker-173K, a large-scale, high-quality training corpus comprising 173k tool-interaction trajectories, explicitly designed to cultivate robust native interleaved tool invocation capabilities.

We conduct extensive evaluations across a diverse suite of long-video understanding and reasoning benchmarks. Experimental results demonstrate that Video-o3 consistently and substantially outperforms the latest MLLMs in this domain. In particular, Video-o3 achieves average accuracies of 72.1% on MLVU, 47.6% on LVBench, and 46.5% on Video-Holmes. These results underscore the robustness and efficiency of Video-o3 in handling long-form video understanding and multi-hop reasoning tasks.

## 2. Related work

**MLLMs for Video Understanding.** While multimodal large language models (MLLMs) (Bai et al., 2025b;a; Team, 2025b;a; Wang et al., 2025d; Ouyang et al., 2025b) have

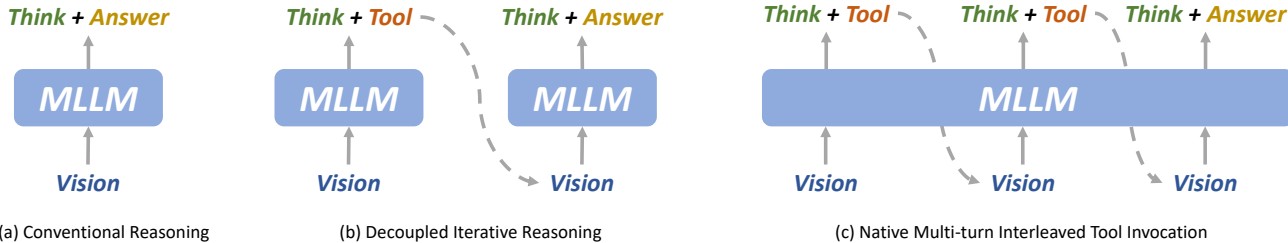

*Figure 2.* **Schematic comparison of different reasoning paradigms.** (a) Conventional Reasoning: The MLLM attempts to derive the answer directly from the initial input without intermediate exploration. (b) Decoupled Iterative Reasoning: The model performs multi-step reasoning via independent calls. Crucially, the context is reset or isolated between turns. (c) Native Interleaved Tool Invocation (Ours): It executes multi-hop reasoning within a unified shared context. This design enables the preservation of both visual features and reasoning history across interleaved turns, facilitating holistic joint inference.

demonstrated impressive capabilities in understanding short video clips, extending these abilities to long-form videos remains challenging (Yan et al., 2025b; Wang et al., 2025c). The primary difficulty lies in balancing comprehensive coverage of temporal information with computational efficiency. Early approaches (Zhang et al., 2024; Zeng et al., 2025b) rely on uniform frame sampling, which inevitably introduces redundancy and may miss critical moments. To reduce such redundancy, keyframe and segment selection methods identify sparse visual evidence or compact memories before reasoning (Tan et al., 2024; Song et al., 2024; Sun et al., 2025; Cao et al., 2025; Wang et al., 2025e). Recent works inspired by reasoning advances in OpenAI o1 (Jaech et al., 2024) and DeepSeek-R1 (Guo et al., 2025) have explored RL-based post-training paradigms. Video-R1 (Feng et al., 2025) and VideoChat-R1 (Li et al., 2025b) employ GRPO (Shao et al., 2024) algorithms with multi-task reasoning data, achieving notable improvements in temporal grounding and video QA. However, these methods (Chen et al., 2025; Wang et al., 2025a; Meng et al., 2025; Zhang et al., 2025a) remain limited to text-based chain-of-thought reasoning with passively provided visual inputs, lacking the ability to actively seek task-relevant evidence.

**Tool-Augmented MLLMs.** Augmenting LLMs with external tools (Team et al., 2025b;a; Sun et al., 2024; Zhao et al., 2025b) has proven effective in expanding model capabilities beyond pure text generation. This paradigm has recently been extended to multimodal scenarios. OpenAI o3 (Team, 2025c) pioneered the "think with images" approach, where models actively gather visual information through operations like zooming in and perform interleaved image-text reasoning. Follow-up works (Zheng et al., 2025; Liu et al., 2025b; Lai et al., 2025; Hong et al., 2025) have further explored various visual tools and interaction patterns. Extending this to video understanding, several recent efforts have investigated "think with videos" paradigms (Zhang et al., 2025b; Li et al., 2025a), and an emerging "clue localization–question answering" pipeline has been explored (Fu et al., 2025b). VideoChat-R1.5 (Yan et al., 2025a) and Video-RTS (Wang et al., 2025f) construct test-time scal-

ing (Snell et al., 2024) systems that iteratively collect evidence through multi-turn tool calls. Native multi-turn approaches, such as Conan (Ouyang et al., 2025c) and Video-Zoomer (Ding et al., 2025), further demonstrate the promise of allowing models to interact with long videos through localized observations. However, these approaches do not explicitly consider the heterogeneous patterns of tool invocation and answer reasoning, and they lack verifiable reward signals to assess the quality of tool-use trajectories. Video-o3 advances these methods in two key respects: decoupling clue-seeking from answer reasoning to mitigate interference between heterogeneous patterns, and introducing verifiable reward signals over tool-use trajectories, enabling precise temporal grounding and effective early termination.

## 3. Methodology

In this section, we introduce Video-o3, a unified framework engineered to facilitate precise clue localization and multi-hop reasoning in long videos through native interleaved multi-turn interactions. By leveraging visual context, Video-o3 dynamically orchestrates tool invocations, utilizing VideoCrop to inspect target segments with adaptive spatiotemporal resolution. Once the accumulated evidence is deemed sufficient to address the query, Video-o3 immediately terminates the search process and derives the final answer by synthesizing information across the multiple retrieved clue segments.

### 3.1. Overall Architecture

The architectural overview of Video-o3 is depicted in Figure 2. In the initial interaction, the model is provided with tool-usage instructions, the user query, and a global view of the video. Upon processing these inputs, the model engages in an internal reasoning process: it decomposes the query to pinpoint visual evidence and evaluates the sufficiency of the current observation. This evaluation drives the model to adopt one of two distinct strategies: (1) **Clue Seeking**: If the available clues are ambiguous or insufficient, the model invokes a tool to scrutinize the granular details of a

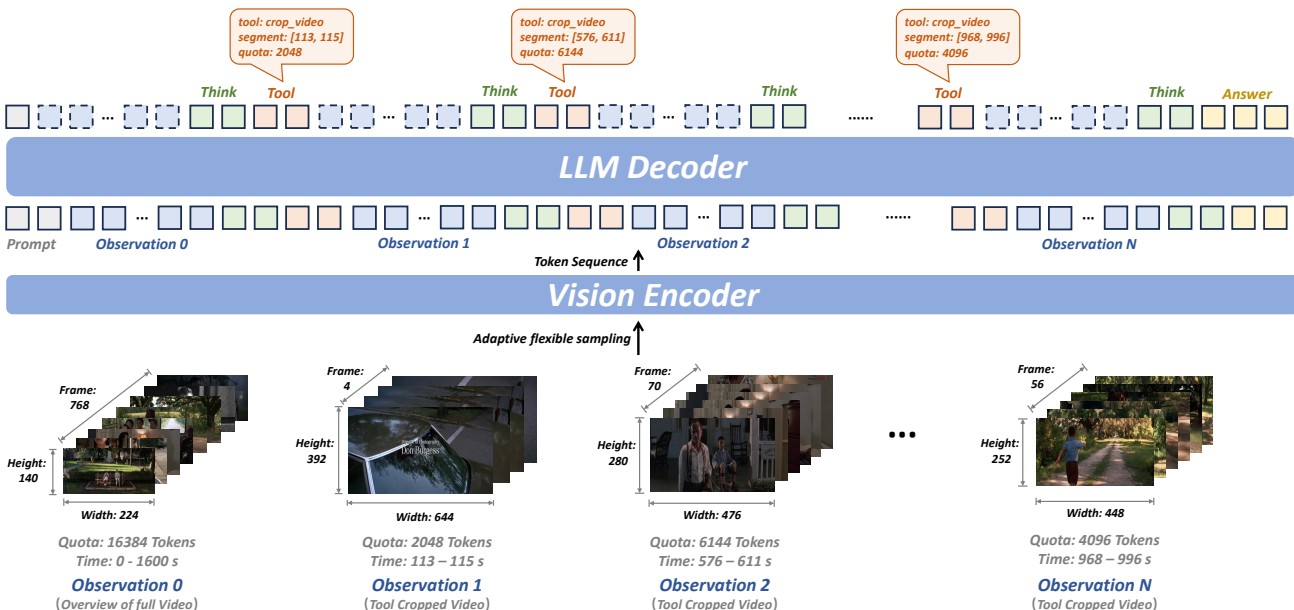

*Figure 3.* **Overview of our proposed Video-o3.** It dynamically executes tool invocations based on previously reasoning to scrutinize specific clue segments via VideoCrop with adaptive spatiotemporal resolution. When the accumulated clues are sufficient to answer the question, Video-o3 is able to terminate the search process and combine multiple clue segments to derive the correct answer.

specific video segment, thereby resolving the uncertainty. (2) **Answer Reasoning**: If clear visual evidence is identified to substantiate the answer, the model generates the final response directly.

Upon opting for clue seeking, the model generates a structured directive containing the temporal window and the visual token quota for the current turn, which guides the external tool in extracting the target video segment. The external tool system dynamically calculates token limit per frame according to the visual quota (see Appendix A for specific calculation formulations). The resampled clip is subsequently integrated with the prompt into the conversation, triggering the next phase of inference. This feedback loop recurs until the model converges on a final answer.

### 3.2. Task-Decoupled Cold-Start

While the shared-context architecture enables synergy between step-wise actions and end-to-end model optimization, it also introduces a critical attention dispersion issue. The heterogeneous context buffer interleaves low-resolution global video tokens, tool-derived fine-grained local fragments, and intermediate reasoning text, causing all tokens to share a full receptive field regardless of task relevance. As a result, attention may be distracted by irrelevant context. For instance, during clue-seeking steps, attention can be diverted by previously cropped video segments when the full video context is required. Similarly, in the answering stage, we observe fake thinking: despite successfully retrieving evidence, the final prediction is inconsistent with the

intermediate reasoning (see Appendix I). This phenomenon mirrors faithfulness issues previously reported in purely textual LLM reasoning (Lyu et al., 2023).

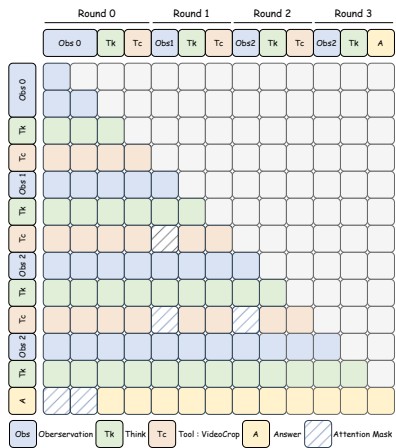

*Figure 4.* **Illustration of Task-Decoupled Attention Masking.** The heatmap illustrates the visibility constraints imposed during the multi-turn supervised fine-tuning process. The global overview is masked during the Answer phase to prevent fake thinking, while local tool outputs are masked during Tool Call planning to force reliance on global view.

To address this, we introduce **Task-Decoupled Attention Masking (TDAM)** for efficient SFT, as illustrated in Figure 4. This strategy explicitly disentangles clue localization from answer reasoning by enforcing strict visibility constraints during the SFT process, effectively isolating the training of these two modes. Specifically, during the clue seeking phase, the model is restricted to attending solely

to the global video input, compelling it to learn planning strategies based on global context. Conversely, during the answer reasoning phase, the global overview is masked, forcing the model to derive answers exclusively from the high-resolution tool observations. To strike a balance between this decoupled expertise and the need for holistic reasoning, we apply this masking to only 10% of the tool-use training data. This ensures that the model retains the capability to simultaneously synthesize global context and fine-grained details. By simulating the efficacy of decoupled expert systems within a unified framework, our approach enables robust adaptation to the complex paradigm of multi-clue localization and multi-hop reasoning.

Formally, let $\mathcal{V}_g$ denote the set of visual tokens representing the global observation, and $\mathcal{V}_l$ denote the set of visual tokens derived from subsequent tool invocations. Let $i$ be the index of the token currently being generated, and $j$ be the index of the context token. We define the task-decoupled attention mask $\mathbf{M}_{i,j}$ as follows:

$$\mathbf{M}_{i,j} = \begin{cases} -\infty, & \text{if } \mathrm{S}(i) = \textsc{Tool} \text{ and } j \in \mathcal{V}_l , \\ -\infty, & \text{if } \mathrm{S}(i) = \textsc{Answer} \text{ and } j \in \mathcal{V}_g , \\ 0, & \text{otherwise} . \end{cases} \quad (1)$$

where $\mathrm{S}(i)$ indicates the current strategies of the model.

### 3.3. Trajectory-Guided Reinforcement Learning

Each tool invocation entails detailed observations of local video segments, which inherently incur substantial token consumption. This often leads to context-length overflow and excessive computational costs. We address this challenge from two complementary perspectives. First, we emphasize **precise clue localization**, requiring the model to accurately identify only the most relevant video segments, thereby minimizing context waste from irrelevant retrievals. Second, we promote **proactive exploration termination**, enabling the model to assess whether the accumulated evidence is sufficient for the given query and to halt further exploration accordingly. This prevents unnecessary tool interactions and significantly reduces redundant computational overhead.

To achieve this, we introduce a **Verifiable Trajectory-Guided Reward (VTGR)**. This mechanism is designed to strike a delicate balance between unconstrained autonomous exploration and efficiency-driven trajectory regularization. Specifically, we formulate the reward function $R$ as a composite of answer correctness, structural validity, and exploration efficiency:

$$R = r_a \cdot (1 + \beta) + r_f , \quad (2)$$

where $r_a \in \{0, 1\}$ represents the base answer reward, and $r_f \in [0, 1]$ denotes the format reward, defined as the normalized ratio of valid formatting across all turns. The term

$\beta$ is the core **trajectory-guided multiplier**, which dynamically modulates the answer reward based on localization precision and path conciseness:

$$\beta = (b_0 + w_g \mathcal{S}_{\text{clue}}) \cdot \gamma . \quad (3)$$

Here, $b_0$ serves as a base additional bonus, and $w_g$ is a weight coefficient controlled as a hyperparameter. The term $\mathcal{S}_{\text{clue}}$ (Hybrid Clue Score) incentivizes *precise localization*, while $\gamma$ (Turn Decay Factor) promotes *agile termination*. Intuitively, $\beta$ grants trajectory-level bonus only through the answer reward: when the answer is correct, the bonus becomes larger if the predicted grounding intervals better align with reference clues and the trajectory terminates with fewer redundant turns.

**Hybrid Clue Score:** To mitigate context waste from erroneous seeking, we employ a tag-based strategy to guide the precision of clue seeking. We categorize samples with a tag $\tau$. For "free exploration" samples ($\tau = \text{free}$), $\mathcal{S}_{\text{clue}}$ is set to a constant $C_{\text{free}}$ to encourage diversity without requiring intermediate trajectory annotations. Conversely, for "trajectory guided" samples ($\tau = \text{tg}$), the score is derived from the alignment between the predicted interval and the ground truth. This dynamically adjusts the bonus based on clue localization precision, discouraging the model from wasting the context window on irrelevant segments. Specifically, the Hybrid Clue Score is calculated as follows:

$$\mathcal{S}_{\text{clue}} = \begin{cases} C_{\text{free}}, & \text{if } \tau = \text{free} \\ \frac{2 \cdot \text{IoU} + \text{IoP} + \text{IoG}}{4}, & \text{if } \tau = \text{tg} \end{cases} , \quad (4)$$

where IoU, IoP, and IoG represent the Intersection over Union, the Intersection over Prediction, and the Intersection over Ground Truth, respectively.

**Turn Decay Factor:** To ensure agile termination and prevent redundant loops, we apply an over-turn penalty. Let $k_t$ denote the actual number of tool invocations, and $k_{ref}$ be the annotated reference limit. The decay factor $\gamma$ penalizes trajectories that exceed the necessary steps:

$$\gamma = \begin{cases} 1 - \lambda \cdot (k_t - k_{ref}), & \text{if } k_t > k_{ref} \\ 1, & \text{otherwise} \end{cases} , \quad (5)$$

where $\lambda$ is the decay penalty weight. This mechanism effectively discourages the model from engaging in meaningless tool calling when evidence is already sufficient, promoting a concise reasoning chain.

**Optimization:** Once the reward scores are established, we optimize the policy with DAPO (Yu et al., 2025b), a GRPO (Shao et al., 2024) variant that refines the policy by maximizing relative advantages within each sampled group. To further stabilize training specifically for long-form interactions, we adopt the over-turn masking introduced in Mini-o3 (Lai et al., 2025). Please refer to Appendix B for detailed implementation specifics.

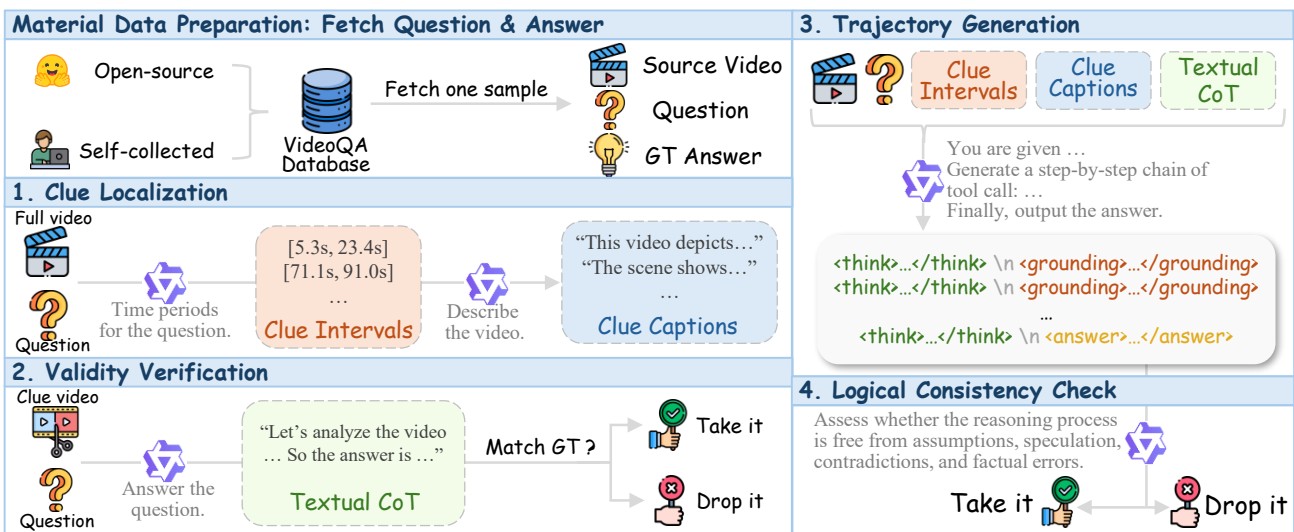

*Figure 5.* **Overview of the data construction pipeline.** We initiate the process by curating high-quality "Video-Question-Answer" database. These inputs are then transformed into explicit tool exploration trajectories via a rigorous four-stage annotation pipeline. Human verification is enforced through random sampling in all stages.

## 4. Dataset

Training MLLMs to master native interleaved tool invocation is severely hindered by the paucity of high quality data with exploration trajectories. Prevailing long-video datasets are predominantly restricted to static "Video-Question-Answer" triplets, lacking explicit, timestamp-anchored intermediate reasoning chains. This absence makes it difficult for models to learn complex multi-step exploration behaviors through supervised paradigms. To bridge this gap, we introduce a scalable, automated data synthesis pipeline capable of synthesizing large-scale training data for both Supervised Fine-Tuning and Reinforcement Learning. Using this pipeline, we construct Seeker-173K, a premium dataset composed of native multi-turn tool interaction trajectories aimed at cultivating efficient and autonomous tool-use capabilities.

**Data Synthesis Pipeline.** We collect existing premium "Video-Question-Answer" triplets and supplement them with high-quality long-video data newly constructed via Gemini 2.5 Pro. Serving as the raw corpus, these samples are processed through a rigorous four-stage pipeline to generate verifiable tool invocation trajectories: *(1) Clue Localization*: We first feed the raw triplets into a VLM to identify all temporal segments containing critical visual cues and to generate detailed descriptions for each interval. *(2) Validity Verification*: To eliminate noise, we extract these localized clips and re-evaluate them with the VLM against the original query. We retain only those samples where the ground-truth answer can be correctly derived solely from the cropped segments, thereby guaranteeing the sufficiency of the visual evidence. *(3) Trajectory Generation*: We then feed the question, verified segments, and descriptions into a

powerful VLM. The model is instructed to synthesize a step-by-step Chain-of-Thought with formatted tool invocations, producing explicit intermediate trajectories anchored by precise timestamps. *(4) Logical Consistency Check*: Finally, an LLM acts as a verifier to scrutinize the generated logical chains. We rigorously filter out flawed instances, preserving only those exhibiting sound logic, rigorous reasoning, and strong support from factual visual evidence.

**The Seeker-173K Dataset.** Leveraging this pipeline, we curate Seeker-173K, a large-scale dataset comprising approximately 173k high-quality trajectories explicitly designed to instill adaptive agentic behaviors. Seeker-173K is rigorously stratified into a four-quadrant taxonomy based on evidence cardinality and visual saliency. This structured diversity enables the model to master distinct capabilities: *(1) Adaptive Invocation*, where the model learns to bypass tool usage when global information is sufficient and to deploy tools only when clues are subtle or fleeting. *(2) Complex Reasoning*, where the model performs logical chaining across disparate timestamps. Furthermore, to enhance robustness, we incorporate supplementary Self-Reflection and Free-Format tasks, providing supervision on error correction and autonomous planning. For comprehensive dataset statistics and task definitions, please refer to Appendix C.

## 5. Experiments

### 5.1. Implementation Details

**Supervised Fine-Tuning**. We initialize our model with Qwen2.5-VL-7B-Instruct (Bai et al., 2025b). Training is conducted in two stages. In the first stage, we train for one

*Table 1.* **Comparison of our method with existing approaches on video question answering tasks across various benchmarks.** Results without special marks are directly taken from the corresponding papers, the "-" indicates that the original paper does not report the result, and * denotes our reproduced result under the same evaluation environment and initial visual-token quota (16k) as Video-o3. The best performance on each benchmark is highlighted in bold.

| Methods | Sizes | VideoMME | MLVU | LVBench | LongVideoBench | VideoMMMU | MMVU | Video-Holmes | MMR-V |
|---|---|---|---|---|---|---|---|---|---|
| | | Avg | M-Avg | Avg | Avg | Overall | M-Avg | Avg | Overall |
| *Open-source Single-Turn Video MLLMs* | | | | | | | | | |
| Qwen2.5-VL (Bai et al., 2025b) | 7B | 65.1 | 70.2 | 45.3 | 56.0 | 47.4 | 61.3 | 34.7 | 32.4 |
| LLaVA-Video (Zhang et al., 2024) | 7B | 63.3 | 70.8 | - | 58.2 | - | - | - | 17.6 |
| Video-R1 (Feng et al., 2025) | 7B | 61.4 | - | - | - | 52.4 | 63.8 | 38.5 | 36.3 |
| VideoChat-R1 (Li et al., 2025b) | 7B | 63.6 | 68.7 | - | 58.2 | - | - | 37.5 | 36.1 |
| Rewatch-R1 (Zhang et al., 2025a) | 7B | 65.6 | - | 43.3 | - | 51.9 | - | 44.3 | - |
| Video-Thinker (Wang et al., 2025b) | 7B | - | - | 37.0 | - | - | - | 43.2 | - |
| *Open-source Decoupled Iterative Reasoning Video MLLMs* | | | | | | | | | |
| Video-RTS (Wang et al., 2025f) | 7B | 63.0 | - | - | 56.6 | **52.7** | 66.4 | - | - |
| Video-MTR (Xie et al., 2025) | 7B | 59.0 | 59.7 | 38.6 | 56.4 | - | - | 35.7 | 36.5 |
| LOVE-R1 (Fu et al., 2025b) | 7B | 64.3* | 69.9* | 46.7* | 58.0* | 50.6* | 65.0* | 39.5* | 43.9* |
| *Open-source Native Multi-turn Tool Invocation Video MLLMs* | | | | | | | | | |
| Conan (Ouyang et al., 2025c) | 7B | 60.5 | 63.4 | 39.2 | 56.6 | - | - | 44.6 | 42.7 |
| LongVT (Yang et al., 2025) | 7B | 64.3 | - | 41.3 | - | 45.4 | - | - | - |
| Video-Zoomer (Ding et al., 2025) | 7B | 64.6* | 69.9* | 44.0* | 55.9* | 51.4* | 65.6* | 43.8* | 39.5* |
| Video-o3 (RL) | 7B | 66.1 | 71.9 | 47.5 | 59.3 | 50.0 | 66.9 | 46.1 | **45.3** |
| Video-o3 (SFT+RL) | 7B | **66.5** | **72.1** | **47.6** | **60.5** | 51.7 | **67.2** | **46.5** | 44.7 |

epoch on approximately 418k cold-start trajectories, covering a diverse set of task types, including multiple-choice questions, open-ended questions, captioning, and grounding. In the second stage, we train on approximately 150k examples for one epoch, of which 100k are sampled from the first-stage data, and we additionally include 50k tool-use examples to elicit the model's tool invocation capabilities. We set the learning rate to $1 \times 10^{-5}$ and the global batch size to 256, with a hard cap on the visual context window at 32k tokens.

**Reinforcement Learning.** Adhering to the protocols in Mini-o3 (Lai et al., 2025), we employ the DAPO algorithm (Yu et al., 2025b) for policy optimization. To enhance training stability, we integrate clip-higher, dynamic sampling, and token-level policy loss. The hyperparameters are configured as follows: a group size of 16, a clip ratio of 0.20, and a batch size of 32. We utilize a constant learning rate of $1 \times 10^{-6}$ and explicitly forgo KL or entropy regularization. Furthermore, to ensure training efficiency, we limit the maximum reasoning turns to 6 and maintain the maximum visual context budget at 32k tokens.

**Evaluation.** At test time, all videos are sampled at 2 fps, with an upper limit of 768 frames. The model employs deterministic decoding, where the overview quota for the initial input is set to 16k tokens, while the quotas for subsequent tool calls are dynamically controlled by the model. The tool invocation limit is set to 8 rounds. Upon reaching this limit, we inject a specific system prompt to alert the model of budget exhaustion and force the immediate generation of a final answer. The evaluation prompts are provided in Appendix F.

## 5.2. Performance in Video QA

We conduct a rigorous evaluation of Video-o3 across a comprehensive suite of long video understanding (Fu et al., 2025a; Zhou et al., 2025; Wang et al., 2025c; Wu et al., 2024) and reasoning benchmarks (Hu et al., 2025; Zhao et al., 2025a; Cheng et al., 2025; Zhu et al., 2025a). The detailed quantitative results are summarized in Table 1.

**Long Video Understanding.** On VideoMME, our RL-only variant achieves an accuracy of 66.1%, surpassing the leading competitor VideoZoomer (64.6%). This performance is further elevated to 66.5% following SFT cold-start initialization. Notably, Video-o3 exhibits superior capability on benchmarks necessitating precise observation of local details, such as MLVU, LVBench, and LongVideoBench. Even without SFT, the RL-trained model outperforms existing native tool-use methods by a substantial margin, securing accuracies of 71.9%, 47.5%, and 59.3%, respectively. The integration of SFT further solidifies this dominance, attesting to Video-o3's robust capacity for long-context perception.

**Video Reasoning.** Video-o3 demonstrates exceptional proficiency in the domain of complex reasoning. On VideoM-MMU, a benchmark designed to assess multidisciplinary reasoning, the RL-only model achieves a commendable 50.0%, which further improves to 51.7% with SFT initialization. The model's capability is particularly evident on Video-Holmes, a dataset demanding intricate multi-hop clue inference. Here, even the RL-only baseline attains a robust 46.1%, while the SFT-enhanced variant refines this to 46.5%. On MMR-V, which requires reasoning over implicit

visual evidence beyond textual priors, Video-o3 improves over Qwen2.5-VL from 32.4% to 44.7%. These results underscore the efficacy of Video-o3 in disentangling and deducing complex multi-hop visual evidence.

## 5.3. Performance in Temporal Grounding

*Table 2.* **Comparison of our method with existing approaches on temporal grounding task.** Best results are in bold.

| Method | Charades-STA | | | |
|---|---|---|---|---|
| | R@0.3 | R@0.5 | R@0.7 | mIoU |
| TimeSuite | 79.4 | 67.1 | 43.0 | - |
| Qwen2.5-VL | 76.1 | 42.9 | 26.2 | 43.6 |
| LongVT | 41.0 | 25.8 | 11.7 | 27.2 |
| **Video-o3 (Ours)** | **83.3** | **71.9** | **49.0** | **60.7** |

Besides general question answering, we evaluate the capability of Video-o3 in temporal grounding. As a critical meta-capability, precise temporal grounding underpins the success of multi-hop reasoning by ensuring the accurate retrieval of evidence. Table 2 details the performance comparison on the Charades-STA benchmark. Notably, LongVT demonstrates weak localization accuracy, with an mIoU of 27.2, falling short of even the baseline Qwen2.5-VL (mIoU: 43.6). In contrast, Video-o3 delivers robust performance with an mIoU of 60.7. This result substantiates the precision of our method in pinpointing key video segments, a meta-ability that is instrumental in enabling efficient, high-precision multi-turn clue localization and joint reasoning.

## 5.4. Ablations

In this section, we conduct comprehensive ablation studies on the core components of Video-o3 to validate the efficacy of the Task-Decoupled Attention Masking, the Verifiable Trajectory-Guided Reward, and the multi-turn tool invocation mechanism. For additional ablation analyses, please refer to Appendix E.

*Table 3.* **Ablation study on the key components of Task-Decoupled Attention Masking (TDAM).** Best results are in bold. $\Delta$ denotes the improvement of Video-o3 (SFT) over the Baseline.

| Method | LVBench | LongVideoBench | MMVU | Video-Holmes |
|---|---|---|---|---|
| | Avg | Avg | M-Avg | Avg |
| w/o TDAM (Baseline) | 43.1 | 55.0 | 60.6 | 40.5 |
| w/o ans. mask | 42.2 | 57.0 | 61.9 | 39.1 |
| w/o gr. mask | 44.1 | **57.1** | 60.6 | 40.8 |
| **Video-o3 (SFT)** | **44.2** | 56.6 | **64.5** | **41.3** |
| $\Delta$ | +1.1 | +1.6 | +3.9 | +0.8 |

**Effectiveness of Task-Decoupled Attention Masking.** To validate the efficacy of our proposed TDAM, we conduct ablation studies by removing masking components during the SFT phase, with results reported in Table 3. It can be observed that removing the attention masking mechanism en-

tirely (denoted as Baseline) leads to a decline in overall performance across both long-video understanding and reasoning tasks. For instance, the accuracy on LongVideoBench drops from 56.6% to 55.0%. Furthermore, when applying only the answer mask or the grounding mask partially, the model's performance still fails to match the level achieved by the full masking strategy. This evidence suggests that our Task-Decoupled Attention Masking effectively separates the tasks of clue localization and answer reasoning. By preventing interference between these different tasks, the strategy enables the model to converge more efficiently toward the desired pattern of "multi-turn clue seeking + multi-hop answer reasoning."

*Table 4.* **Ablation study on the key components of Verifiable Trajectory-Guided Reward (VTGR).** Best results are in bold. $\Delta$ denotes the improvement of Video-o3 (RL) over the Baseline.

| Method | LVBench | LongVideoBench | MMVU | Video-Holmes |
|---|---|---|---|---|
| | Avg | Avg | M-Avg | Avg |
| w/o VTGR (Baseline) | 46.1 | 56.5 | 62.6 | 42.9 |
| w/o Hybrid Clue Score | 45.2 | 57.3 | 64.7 | 41.8 |
| w/o Turn Decay Factor | 46.7 | 58.9 | 65.0 | 45.5 |
| **Video-o3 (RL)** | **47.5** | **59.3** | **66.9** | **46.1** |
| $\Delta$ | +1.4 | +2.8 | +4.3 | +3.2 |

**Effectiveness of the Verifiable Trajectory-Guided Reward.** We further dissect the impact of each component within VTGR through ablation studies, as detailed in Table 4. The removal of the bonus multiplier causes the unified reward to degenerate into basic correctness and format signals. This reduction fails to stimulate tool-use behaviors in the early stages, rendering the training process unstable and difficult to converge. Without the Hybrid Clue Score, the framework loses critical constraints on the tool invocation process, failing to guide the model toward efficient reasoning trajectories. Furthermore, the absence of the Turn Decay Factor results in uncontrolled expansion of reasoning turns. This often leads to trajectory lengths that violate inference-time caps, causing the model to fail in delivering a final response. Collectively, these results demonstrate that our Verifiable Trajectory-Guided Reward is essential for regularizing the reasoning process: it incentivizes the exploration of precise clue segments while curbing superfluous interactions, ultimately guaranteeing the accuracy and efficiency of multi-hop reasoning.

*Table 5.* **Ablation study on max turn limit and model accuracy.** Best results are in bold.

| Max Turn | Video-MME | MLVU | VideoMMMU | LVBench |
|---|---|---|---|---|
| | Avg | Avg | Avg | Avg |
| 2 | 63.6 | 67.4 | 48.0 | 46.8 |
| 4 | 65.2 | 69.5 | 49.0 | 47.2 |
| **8 (Ours)** | **66.5** | **72.1** | **51.7** | **47.6** |

**Upper Limit on Turns During Testing.** To quantify the contribution of multi-hop reasoning to model performance,

*Table 6.* **Efficiency comparison on MLVU.** We report GPU memory, single-A100 inference time, input visual tokens, response tokens, and accuracy under the same MLVU evaluation setting.

| Model | Inference Type | GPU Mem. (GB) | Time (s) | Input Vis. Tokens | Resp. Tokens | Acc. |
|---|---|---|---|---|---|---|
| VideoChat-R1-7B (Li et al., 2025b) | Single-turn | 18.2 | 6.3 | 14,368 | 104 | 68.7 |
| VideoChat-R1.5-7B (Yan et al., 2025a) | Decoupled Iterative | 18.5 | 18.9 | 41,515 | 343 | 70.9 |
| LOVE-R1-7B (Fu et al., 2025b) | Decoupled Iterative | 18.3 | 15.2 | 32,108 | 270 | 69.9 |
| Video-o3-7B (Ours) | Native Multi-turn | 18.7 | 10.2 | 18,020 | 235 | 72.1 |

we conducted an ablation study on the maximum number of interaction turns, with results detailed in Table 5. By constraining the upper limit to 2, 4, and 8 turns, we simulated scenarios ranging from coarse inspection to comprehensive investigation. Empirical evidence reveals a consistent positive correlation between interaction depth and inference accuracy across all benchmarks. This trend is most pronounced in datasets demanding fine-grained retrieval and intricate logic, such as MLVU and VideoMMMU. This uplift highlights that complex long-video queries often exceed the capacity of shallow-depth inference. The sustained improvement at 8 turns validates the core premise of Video-o3: our Native Interleaved Tool Invocation paradigm empowers the model to decompose sophisticated queries into manageable sub-goals. Rather than succumbing to context drift, Video-o3 leverages the extended budget to iteratively resolve ambiguities, effectively transforming uncertain initial hypotheses into verified conclusions through a robust multi-hop reasoning chain.

### 5.5. Efficiency Analysis

To quantify the efficiency of our native multi-turn tool invocation, we compare GPU memory, inference time, input visual tokens, response tokens, and accuracy on MLVU in Table 6. Video-o3 achieves the highest accuracy of 72.1% with 18.7GB GPU memory and 10.2 seconds per sample. Compared with the single-turn baseline VideoChat-R1, Video-o3 obtains a 3.4% accuracy gain with an acceptable increase in computational overhead. Compared with decoupled iterative reasoning methods, it reduces inference time by 32.9% over LOVE-R1 (15.2 seconds) and 46.0% over VideoChat-R1.5 (18.9 seconds), while using fewer input visual tokens and response tokens than both. This efficiency comes from the unified shared context of Video-o3: unlike decoupled methods that process each turn independently and repeatedly re-encode the multimodal context, our multi-turn reasoning paradigm naturally leverages KV cache to eliminate redundant computation, enabling the model to scale from single-turn to multi-turn reasoning with modest additional cost.

### 5.6. Generalization to Smaller Base Models

To examine whether our native multi-turn tool invocation framework generalizes beyond the Qwen2.5-VL-7B back-

*Table 7.* **Generalization to a smaller base model.** Results are evaluated under a 16k visual-token budget. Best results are in bold.

| Model | VideoMME | MLVU | LVBench | LongVideoBench | Video-Holmes |
|---|---|---|---|---|---|
| | Avg | M-Avg | Avg | Avg | Avg |
| Qwen3-VL-4B | 66.9 | 72.0 | 46.0 | 61.0 | 37.3 |
| Video-o3-4B | **67.3** | **73.2** | **50.0** | **62.5** | **48.7** |
| Δ | **+0.4** | **+1.2** | **+4.0** | **+1.5** | **+11.4** |

bone, we instantiate it on the newer and smaller Qwen3-VL-4B model (Bai et al., 2025a) and evaluate under 16k initial visual-token budget. As shown in Table 7, Video-o3-4B consistently improves over Qwen3-VL-4B across all five benchmarks, with gains of 0.4 on VideoMME, 1.2 on MLVU, 4.0 on LVBench, 1.5 on LongVideoBench, and 11.4 on Video-Holmes. These results show that the framework can transfer to smaller base models, while its final performance still depends on the base model's temporal grounding ability, which provides the foundation for reliable clue localization and multi-hop video reasoning.

## 6. Conclusion

In this work, we have presented Video-o3, a framework empowering MLLMs with native interleaved tool invocation for long-form video understanding. To enable robust end-to-end training, we introduce Task-Decoupled Attention Masking to mitigate task interference and a Verifiable Trajectory-Guided Reward to optimize the trade-off between exploration and efficiency. Supported by our constructed Seeker-173K dataset, Video-o3 establishes new state-of-the-art performance across multiple benchmarks. It underscores the transformative potential of native tool-use paradigms in advancing the next generation of agentic MLLM.

## Acknowledgements

This work is supported by the National Key R&D Program of China (No. 2022ZD0160900), the Basic Research Program of Jiangsu (No. BK20250009), the Fundamental Research Funds for the Central Universities (No. 020214380140), the Fundamental and Interdisciplinary Disciplines Breakthrough Plan of the Ministry of Education of China (No. JYB2025XDXM118), and the Collaborative Innovation Center of Novel Software Technology and Industrialization.

## Impact Statement

This paper presents Video-o3, a framework designed to advance MLLMs in understanding long-form videos through native interleaved clue seeking. Our work primarily aims to enhance the efficiency of information extraction from large-scale video data, which has significant potential benefits for applications in digital archives, educational content analysis, and personal media management. However, we acknowledge that the capability to autonomously scrutinize fine-grained visual details and perform precise temporal grounding could be repurposed for unintended surveillance or privacy-intrusive monitoring. While our research focuses on general-purpose video reasoning using public datasets and publicly accessible web videos, we emphasize the importance of respecting source licenses, platform terms, and data privacy requirements when constructing or releasing derived annotations. In particular, the self-collected YouTube videos used in our dataset are sourced from public channels under the CC-BY license and are used solely for academic research; we do not extract or store personally identifiable information from these videos. We further encourage strict safeguards and ethical guidelines when deploying such autonomous agents in real-world scenarios, particularly those involving sensitive personal data. We are committed to the responsible development of multimodal AI and encourage the community to prioritize privacy protection in future applications of this technology.

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

# Appendix Overview

## A. Dynamic Visual Quota for Tool Invocation

Here, we elaborate on the design of the dynamic visual token quota for tool invocation. Let $k$ denote the current interaction turn. The structured instruction generated by the model comprises two critical components: (1) The target temporal segment, denoted as "temporal_segment": $[t_{s,k}, t_{e,k}]$, which specifies the start and end timestamps of the video interval intended for scrutiny in the current turn; (2) The visual token budget, denoted as "sampling_strategy": $q_k$. This variable takes values from a discrete set of three predefined granularities (i.e., {'coarse', 'medium', 'fine'}), where each level corresponds to a predefined upper limit on the total token count controlled by hyperparameters.

By controlling the target temporal segment and visual token budget, the model dynamically regulates quota allocation during video input. Specifically, constrained by $[t_{s,k}, t_{e,k}]$, $q_k$, and preset FPS, the system computes the token limit per frame as:

$$T_{\text{frame}} = \frac{q_k}{(t_{e,k} - t_{s,k}) \times \text{FPS}} . \tag{6}$$

This formulation follows standard video sampling practice and governs the spatial resolution. This design establishes a budget-aware mechanism within the limited context window, empowering the model to autonomously trade off between temporal duration and spatial fidelity.

## B. Over-turn Masking GRPO

Aligning with the implementation of Mini-o3 (Lai et al., 2025), we incorporate an over-turn masking mechanism to avoid penalizing responses that exceed the turn limit. Specifically, in addition to the standard reward $r$ and advantage $A$, we introduce a binary completion mask $M$, which indicates whether the response successfully terminates within the prescribed turn constraints. We then calculate the masked advantage $A'_i$ as $A'_i = M_i \cdot A_i$. This operation ensures that over-turn trajectories (where $M_i = 0$) do not generate gradient signals, thereby effectively excluding them from the optimization process.

$$\mathcal{J}_{GRPO}^{over-turn}(\theta) =$$

$$\mathbb{E}_{[q \sim \mathcal{D}, \{o_i\}_{i=1}^G \sim \pi_{\theta_{old}}(\cdot|q)]} \frac{1}{\sum_i^G M_i} \sum_{i=1}^G \left( \min \left( \frac{\pi_\theta(o_i|q)}{\pi_{\theta_{old}}(o_i|q)} A_i \cdot M_i, \text{clip}\left( \frac{\pi_\theta(o_i|q)}{\pi_{\theta_{old}}(o_i|q)}, 1 - \epsilon, 1 + \epsilon \right) A_i \cdot M_i \right) \right) , \tag{7}$$

$$A_i = \frac{r_i - mean(\{r_1, r_2, ..., r_G\})}{std(\{r_1, r_2, ..., r_G\})} , \tag{8}$$

$$M_i = \mathbb{1}\{|o_i| <= C_{context}\} \cdot \mathbb{1}\{\text{turn}(o_i) <= C_{turn}\} . \tag{9}$$

## C. Details of Dataset Construction

### C.1. Task Type Definition

To foster adaptive tool invocation capabilities, we stratify our dataset based on the cardinality and visual saliency of the evidence. Formally, we partition the data along two dimensions: Clue Quantity and Visual Saliency. This yields a four-quadrant taxonomy of Core Tasks, which are systematically generated via our proposed data synthesis pipeline: (1) Single-Clue Direct Answering, (2) Single-Clue Tool Invocation, (3) Multi-Clue Direct Answering, and (4) Multi-Clue Tool Invocation. In addition to these synthesized core tasks, we introduce two supplementary tasks by rigorously filtering and cleaning existing high-quality datasets. These are incorporated to enhance robustness: (5) Self-Reflection Tool Invocation and (6) Free-Format Tool Invocation.

**Single-Clue Direct Answering.** When a single pivotal clue remains distinctly visible throughout the video, the global visual information is sufficient. This allows the model to answer directly, obviating the need for tool invocation.

**Single-Clue Tool Invocation.** These queries rely on a single clue that is spatially localized or appears only briefly. Since the global overview's limited frame rate and resolution cannot capture such fine-grained evidence, the model must deploy tools with a dynamic token budget to scrutinize specific high-resolution details.

**Multi-Clue Direct Answering.** In multi-hop reasoning scenarios, clues are dispersed across discrete segments. However, if these clues possess sufficient temporal duration and visual clarity, the model can synthesize the answer solely from the initial global observation.

**Multi-Clue Tool Invocation.** Clues are distributed across distinct intervals, exhibiting parallel or progressive logical dependencies. Furthermore, critical clues may be temporally fleeting or spatially subtle. This necessitates the use of tools to observe these moments with enhanced spatiotemporal resolution and to logically chain the evidence scattered across different sections.

**Self-Reflection Tool Invocation.** In complex searches, initial retrieval attempts often yield irrelevant or noisy segments. This task simulates such failure cases, training the model to critically evaluate the utility of retrieved content. Instead of hallucinating an answer from unrelated visual data, the model learns to explicitly acknowledge the error, reject the false positive, and autonomously initiate a corrective search strategy to locate the true evidence.

**Free-Format Tool Invocation.** This subset is devoid of intermediate trajectory annotations, providing supervision solely based on the final answer. In this setting, the intermediate exploration process remains entirely unconstrained. This forces the model to autonomously formulate and optimize its own search strategies without external interference.

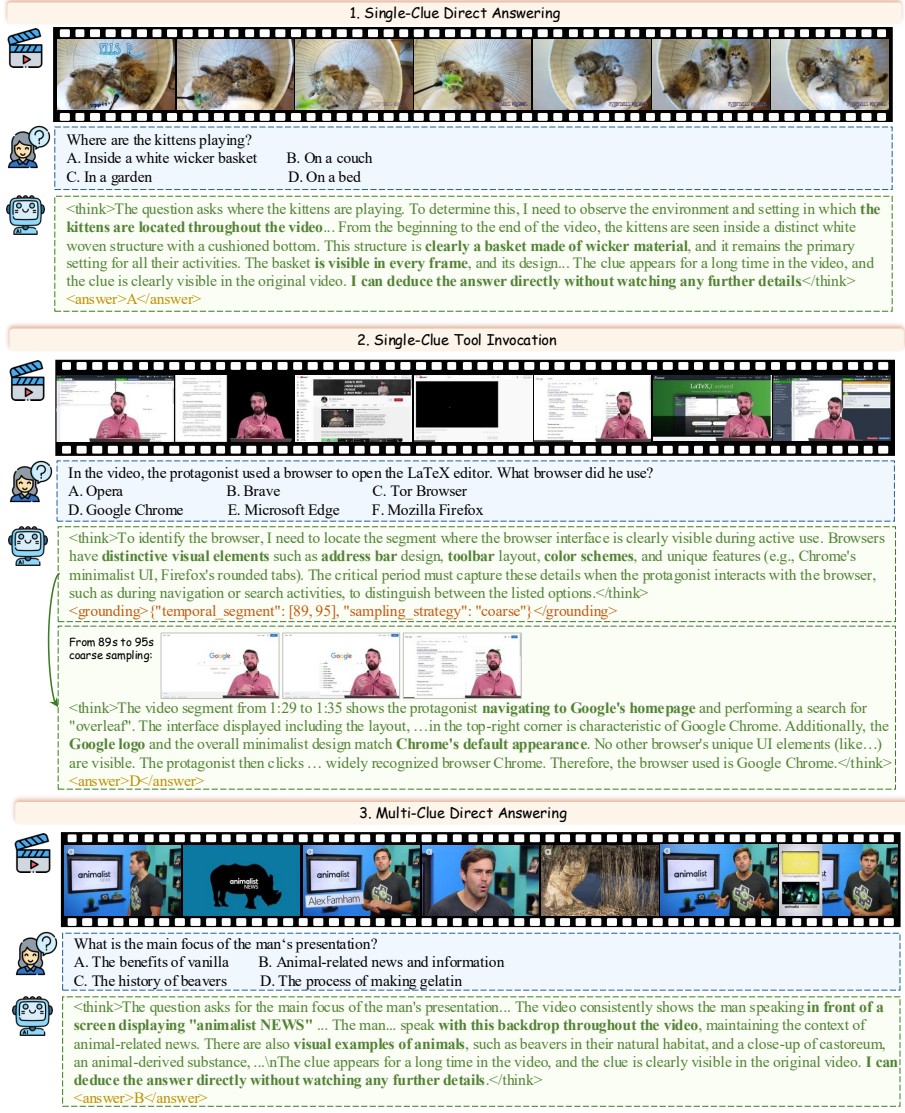

*Figure 6.* **Question–answer demonstrations: single-clue direct answering, single-clue tool invocation, and multi-clue direct answering.**

### C.2. Dataset statistics

As detailed in Table 8, our training dataset is composed of two distinct yet complementary parts: the Conventional set and the Seeker-173K set. The Conventional data constitutes the backbone of our training, featuring 418k samples predominantly

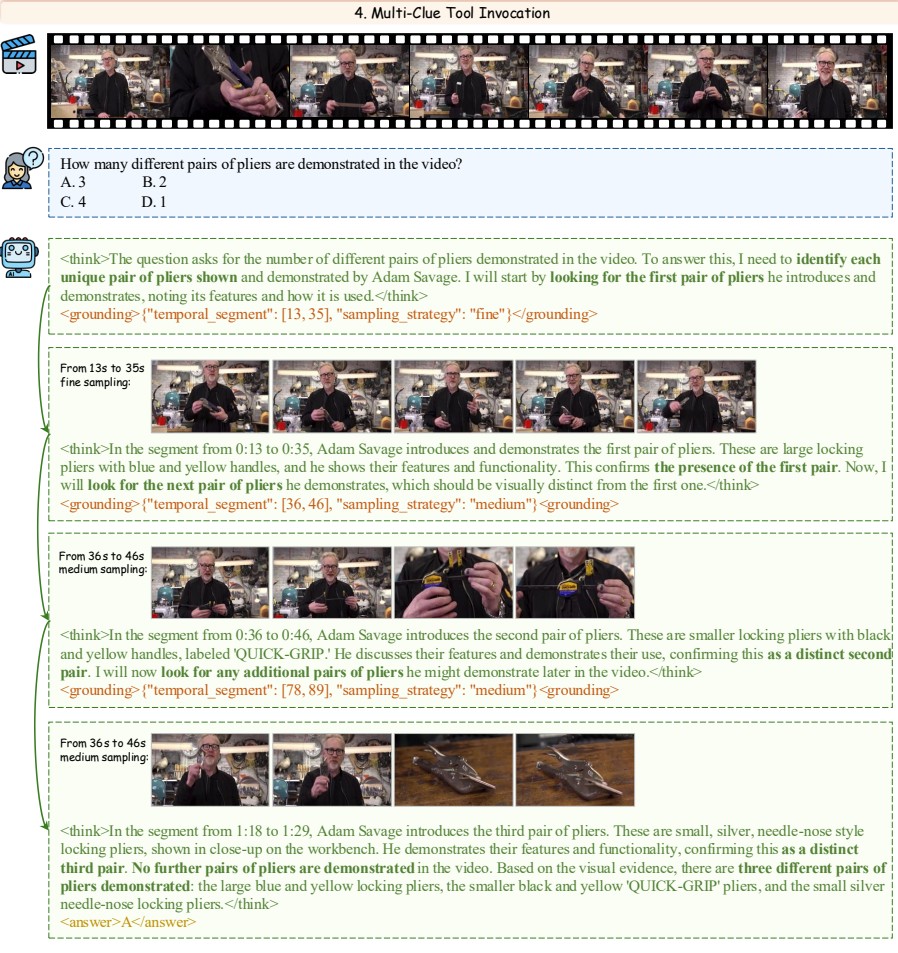

*Figure 7.* **Question–answer demonstrations: multi-clue tool invocation.**

*Table 8.* **Task types and data volumes of Conventional training data and Seeker-173K.**

| Dataset | Task | Source | Instance Num |
|---------|------|--------|--------------|
| *Conventional* | Multiple Choice QA | LLaVA-Video (Zhang et al., 2024) | 168,930 |
| | Open-ended QA | LLaVA-Video (Zhang et al., 2024) | 128,008 |
| | Dense Video Caption | LLaVA-Video (Zhang et al., 2024) | 102,088 |
| | Temporal Grounding | TimeLens (Zhang et al., 2025c) | 19,414 |
| *Seeker-173K* | Single-Clue Direct Answering | LLaVA-Video (Zhang et al., 2024) | 9,946 |
| | Single-Clue Tool Invocation | LLaVA-Video (Zhang et al., 2024) | 79,848 |
| | | CGBench (Chen et al., 2024) | 6,764 |
| | | LongVideoDB | 7,000 |
| | | YouTube (Self-collected) | 13,287 |
| | Multi-Clue Direct Answering | LLaVA-Video (Zhang et al., 2024) | 29,523 |
| | Multi-Clue Tool Invocation | LLaVA-Video (Zhang et al., 2024) | 13,900 |
| | | YouTube (Self-collected) | 1,606 |
| | Self Reflection Tool Invocation | VideoSIAH (Yang et al., 2025) | 2,000 |
| | Free Format Tool Invocation | LongVideo-Reason (Chen et al., 2025) | 9,531 |

sourced from LLaVA-Video (Zhang et al., 2024) and TimeLens (Zhang et al., 2025c). By encompassing fundamental tasks ranging from dense captioning to temporal grounding, this segment ensures the model retains strong general-purpose video comprehension capabilities. The core contribution, Seeker-173K, is strategically curated to instill advanced agentic behaviors. It comprises roughly 173k instances, strictly stratified according to our proposed task definitions. Beyond

repurposing existing datasets like CGBench (Chen et al., 2024) and LongVideoDB, we significantly enrich the dataset with 14.8k self-collected YouTube instances, targeting complex scenarios that necessitate Single-Clue and Multi-Clue tool usage.

## D. Additional Implementation Details

### D.1. Training Details

Our SFT-stage training is built on LLaMA-Factory (Zheng et al., 2024) framework, and our RL-stage training is built on verl (Sheng et al., 2025) framework. During the rollout phase of RL training, we use vLLM (Kwon et al., 2023) to accelerate model inference. We use AdamW (Loshchilov & Hutter, 2017) as the optimizer for both training stages. The key hyperparameters for the different training stages are listed in Table 9 and Table 10.

*Table 9.* **Parameter settings for supervised fine-tuning.**

|  |  |  |
|---|---|---|
| *Data* | Dataset | Conventional + Seeker-173K (Partial) |
|  | # samples (stage1) | 418440 |
|  | # samples (stage2) | 154163 |
| *Vision* | # overview quota | 16384 |
|  | # coarse quota | 2048 |
|  | # medium quota | 4096 |
|  | # fine quota | 6144 |
|  | FPS | 2.0 |
|  | Frame Limit | 768 |
| *Optimize* | Optimizer | AdamW |
|  | Learning rate | 1e-5 |
|  | Learning rate scheduler | cosine |
|  | Global batch size | 256 |
|  | Training epochs | 1 |
|  | GPU Nums | 32 |

*Table 10.* **Parameter settings for reinforcement learning.**

|  |  |  |
|---|---|---|
| *Data* | Dataset | Seeker-173K |
|  | # samples | 12800 (Random sampling) |
| *Vision* | Max Visual Quota | 32768 |
|  | # overview quota | 16384 |
|  | # coarse quota | 2048 |
|  | # medium quota | 4096 |
|  | # fine quota | 6144 |
|  | FPS | 2.0 |
|  | Frame Limit | 768 |
| *Reward* | $b_0$ | 0.5 |
|  | $w_g$ | 0.5 |
|  | $C_{\text{free}}$ | 0.5 |
|  | $\lambda$ | 0.05 |
| *Optimize* | Optimizer | AdamW |
|  | Learning rate | 1e-6 |
|  | Learning rate scheduler | constant |
|  | Global batch size | 32 |
|  | Training steps | 400 |
|  | GPU Nums | 32 |

### D.2. Evaluation Details

Our evaluation is based on VLMEvalKit (Duan et al., 2024), on top of which we implement model inference functions supporting tool invocation and multi-turn reasoning. During evaluation, we set the temperature to 0 and perform greedy decoding to obtain deterministic results. The maximum number of inference rounds is set to 8, and the upper limit of visual tokens is set to 16,384; all other settings follow the same default configuration as Qwen2.5-VL. In addition, consistent with Qwen2.5-VL, we sample videos at 2 FPS with a maximum of 768 frames for all evaluated benchmarks.

# E. Additional Ablations

## E.1. Effectiveness of Different Attention Mask Ratio

*Table 11.* **Ablation study on the mask ratio of Task-Decoupled Attention Masking.**

| Mask Ratio | LVBench | LongVideoBench | MMVU | Video-Holmes | Average |
|---|---|---|---|---|---|
| | Avg | Avg | M-Avg | Avg | Avg |
| 0.3 | **45.2** | 55.9 | 61.0 | 40.6 | 50.7 |
| 0.2 | 43.5 | 56.5 | 60.8 | 40.8 | 50.4 |
| **0.1 (Ours)** | 44.2 | **56.6** | **64.5** | **41.3** | **51.6** |
| 0.0 (No Mask) | 43.1 | 55 | 60.6 | 40.5 | 49.8 |

We further examine the impact of the mask ratio in Task-Decoupled Attention Masking. We selected three mask ratios: 0.1, 0.2, and 0.3, and after SFT, evaluated the performance of different model variants across multiple benchmarks. The results are shown in Table 11. Increasing the mask ratio does not lead to significant performance improvements; on the contrary, it negatively affects the model's performance in complex clue reasoning scenarios (e.g., Video-Holmes). Combined with Table 3 in the main text, this demonstrates that choosing a mask ratio of 0.1 is a highly effective choice, sufficient to alleviate the attention diffusion problem and enhance reasoning capability.

## E.2. Effectiveness of Verifiable Trajectory-Guided Reward

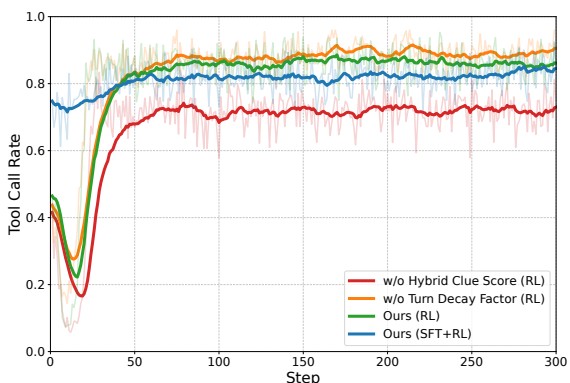

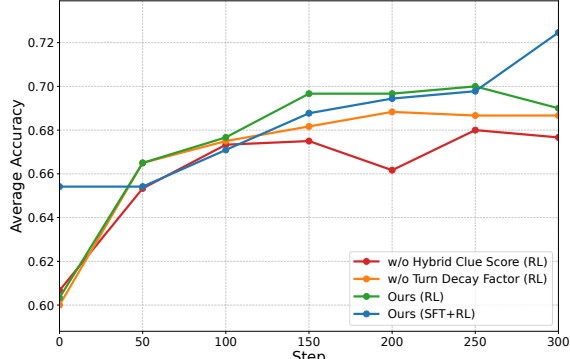

*Figure 8.* **Trend of effective tool-call rates for each sampling instance across different training steps.** A time-weighted exponential moving average (EMA) with a smoothing coefficient of 0.9 is applied to mitigate short-term fluctuations and highlight the underlying trends.

*Figure 9.* **Accuracy of the model on the validation set across different training steps.** We construct a validation subset of 1,000 samples for efficient evaluation, consisting of 600 samples randomly selected from VideoMME and 400 samples randomly selected from MLVU.

To dissect the influence of our Verifiable Trajectory-Guided Reward on tool-use behavior and performance, we ablate the core elements of the bonus multiplier: the Hybrid Clue Score and the Turn Decay Factor. Figure 8 plots the evolution of effective tool-call rates, while Figure 9 tracks the validation accuracy.

**Role of Hybrid Clue Score:** The removal of the Hybrid Clue Score precipitates a drastic drop in both tool-call frequency and accuracy. Lacking temporal supervision, the model fails to localize pivotal clues, rendering tool invocations ineffective and discouraging their use. More critically, this leads to reward ambiguity: the objective function degenerates into a pure correctness check, where spurious trajectories (correct answer, wrong path) are rewarded equally with valid ones. This lack of process constraint significantly aggravates hallucination.

**Role of Turn Decay Factor:** Excluding the Turn Decay Factor results in a slightly elevated tool-call rate but compromises accuracy compared to Ours (RL). This counter-intuitive result arises because indiscriminate tool invocation on tasks requiring global semantic synthesis can fragment the context, obscuring holistic understanding. Ideally, models should bypass tool use when global evidence is sufficient. The Turn Decay Factor acts as a critical regularizer, guiding the model to adopt distinct strategies (direct answering or iterative seeking) based on the query type, thus optimizing the trade-off between inspection depth and global coherence.

**Necessity of SFT Cold Start:** Notably, the non-SFT model, i.e. Ours (RL), exhibits a "dip-and-recover" pattern in tool

usage: a sharp decline within the first 20 steps followed by a resurgence. We attribute this to the initial incompetence of the vanilla Qwen2.5-VL model; early tool calls are noisy and detrimental to accuracy, prompting the policy to suppress them. However, as the model internally acquires tool semantics, the behavior shifts from "noise" to "signal," triggering a rapid recovery in usage frequency. Conversely, the stability of Ours (SFT+RL) underscores the vital role of SFT initialization in bypassing this unstable exploration phase and establishing robust tool-use capabilities from the outset.

### E.3. Hybrid Clue Score and Temporal Grounding

*Table 12.* **Direct impact of the Hybrid Clue Score on temporal grounding.** We report Charades-STA mIoU under different uses of the Hybrid Clue Score and temporal grounding data.

| Hybrid Clue Score | Temporal Grounding Data | mIoU | $\Delta$ from HCS |
|:---:|:---:|:---:|:---:|
| – | – | 46.5 | – |
| ✓ | – | 53.8 | **+7.3** |
| – | ✓ | 59.6 | – |
| ✓ | ✓ | **60.7** | **+1.1** |

To directly evaluate how the Hybrid Clue Score affects localization precision, we conduct an additional ablation on the temporal grounding benchmark Charades-STA. As shown in Table 12, adding the Hybrid Clue Score improves mIoU from 46.5 to 53.8 without temporal grounding data, and from 59.6 to 60.7 when such data is included. The larger gain in the former setting indicates that the Hybrid Clue Score provides direct process-level supervision for locating pivotal clues, rather than merely encouraging more frequent tool invocation.

### E.4. Turn Distribution Analysis

*Table 13.* **Actual turn distribution under the 8-turn inference limit.** We report the average number of tool-invocation turns and the percentage of samples requiring more than 2, 4, and 6 turns.

| Model Variant | Avg. Turns | >2 Turns | >4 Turns | >6 Turns |
|:---|:---:|:---:|:---:|:---:|
| Video-o3 (SFT) | 2.49 | 24.4% | 16.3% | 2.7% |
| Video-o3 (SFT+RL w/o TDF) | 2.57 | 17.6% | 10.4% | 4.8% |
| **Video-o3 (SFT+RL w/ TDF)** | **2.31** | **15.4%** | **9.9%** | **1.3%** |

We report the actual turn distribution under the 8-turn inference limit in Table 13. With the Turn Decay Factor, Video-o3 reduces the average number of turns from 2.57 to 2.31 compared with the variant without TDF, and the proportion of long trajectories requiring more than 6 turns drops from 4.8% to 1.3%. Meanwhile, only 9.9% of samples require more than 4 turns and 1.3% require more than 6 turns, indicating that the 8-turn limit covers most evaluation cases while leaving enough budget for genuinely complex multi-hop queries.

### E.5. Effectiveness of Training Stage

*Table 14.* **Performance comparison of models after different training stages.**

| Method | LVBench | LongVideoBench | VideoMMMU | Video-Holmes |
|:---|:---:|:---:|:---:|:---:|
| | Avg | Avg | Overall | Avg |
| Qwen2.5-VL (Baseline) | 45.3 | 56.0 | 47.4 | 35.6 |
| Qwen2.5-VL (Tool ZS) | 42.8 | 53.8 | 45.6 | 38.3 |
| Video-o3 (SFT) | 44.2 | 56.6 | 47.9 | 41.3 |
| Video-o3 (RL) | 47.5 | 59.3 | 50.0 | 46.1 |
| **Video-o3 (SFT+RL)** | **47.6** | **60.5** | **51.7** | **46.5** |

To validate the contribution of each training phase, we analyze the performance trajectory across varying training stages, with results summarized in Table 14. The experiments reveal that directly applying multi-turn tool-use prompts to the off-the-shelf Qwen2.5-VL (baseline) precipitates a significant performance degradation compared to direct answering. Moreover, the model struggles to break free from text-only generation patterns, indicating a fundamental lack of inherent awareness for native tool invocation. The introduction of SFT serves as a critical stabilization step, restoring accuracy to baseline levels and even yielding marginal superiority on certain benchmarks (e.g., Video-Holmes sees a 5.7% gain, rising from 35.6% to 41.3%). Implementing RL in isolation allows the model to comprehensively outperform the baseline.

However, in the absence of proper cold-start initialization, interactions are predominantly restricted to simple clue seeking, which prevents the acquisition of complex multi-hop reasoning skills. Ultimately, the synergistic combination of SFT cold-start and RL post-training delivers optimal performance. This pipeline effectively enables the model to master native interleaved tool invocation, facilitating precise multi-hop clue localization and holistic reasoning within long videos.

### E.6. Scalability with Free-Format Data

*Table 15.* **Scalability with sparse trajectory annotations and free-format data.** We report RL results when only a small fraction of samples contain trajectory annotations. Free data denotes Video-QA triplets without intermediate tool-use trajectory annotations. Best results are in bold.

| Method | Trajectory Data | Free Data | VideoMME | MLVU | LVBench |
|---|---|---|---|---|---|
| | | | Avg | M-Avg | Avg |
| 15% trajectory (RL) | 15% | – | 63.8 | 70.2 | 43.7 |
| 15% trajectory + 85% free (RL) | 15% | 85% | 65.6 | **71.9** | 46.8 |
| Video-o3 (RL) | 100% | – | **66.1** | **71.9** | **47.5** |

We further examine whether Video-o3 can scale when only a small subset of training samples contains tool-use trajectory annotations. As shown in Table 15, using only 15% trajectory-annotated data already yields a functional RL model, but its performance remains below the full RL setting. Adding 85% free-format Video-QA data, which contains no intermediate trajectory annotations and only provides final-answer supervision, improves VideoMME, MLVU, and LVBench by 1.8, 1.7, and 3.1 points, respectively. This recovers most of the gap to the fully trajectory-annotated RL model, indicating that dense trajectory annotation is helpful for fast convergence but not strictly required for scaling: free-format data can still encourage autonomous tool strategy optimization through answer-level verifiable feedback.

## F. Prompt Template

Figure 10 demonstrates all prompts employed for evaluation on Video QA benchmarks. The system prompt and user question prompt initially provide detailed instructions guiding the model through reasoning and tool invocation (Zhu et al., 2024a;c). Throughout the iterative process, cropped videos returned by tool calls are wrapped with the prompt for tool calling response and returned to the model as new input. The model is allowed to autonomously determine when to stop reasoning and provide the final answer. However, if the maximum number of tool invocation rounds is reached without conclusion, the prompt for forcing answer is provided to force the model to synthesize the available information and produce the final answer.

**System Prompt for Video QA**

You are a helpful assistant. Answer the user's multiple-choice question based on the provided video.
Output your thinking process within the `<think>` and `</think>` tags.
If you find any video segments that might help answer your questions, you can view a specific area in detail by outputting `<grounding>{\"temporal_segment\": [t0, t1], \"sampling_strategy\": \"medium\"}</grounding>`,
where t0 and t1 are the start and end times (in integer seconds) of the video segment you want to observe in detail within the entire video, sampling_strategy must be a string and in ['coarse', 'fine', 'medium'].
Once you believe you have observed all the video clues that help you answer the question and are able to integrate all the existing clues to
give the correct answer, you should put the correct answer within the `<answer>` and `</answer>` tags.
The following is an example of the thinking process and the final answer:
 - When you want to examine a specific clue segment of a video more closely, produce exactly: '`<think>`*your thinking process here*`</think>`\n`<grounding>`*{"temporal_segment": [START_TIMESTAMP, END_TIMESTAMP], "sampling_strategy": "medium"}*`</grounding>`\n'.
 - When you believe the current information is sufficient to give an answer, produce exactly: '`<think>`*your thinking process here*`</think>`\n`<answer>`*Option Letter*`</answer>`\n'.

**User Question Prompt for Video QA**

Here is the original full video (Observation 0):
`<video>`This video is uniformly sampled at {sample_fps:.2f} fps, contains {total_frames:.1f} frames from 0 seconds to {max_duration:.1f} seconds.
Answer the following question according to the content of the video:
{question}
Options:
{options}
You are advised to first observe potential clue segments, use '`<think>`*YOUR THINK*`</think>`\n`<grounding>`*YOUR GROUNDING*`</grounding>`\n' to to specify the segment you want to observe in detail.
If the evidence is visible in the original video for a long enough time and is clear enoug

**Prompt for Tool Calling Response**

After the above Action {action_turn}, here is the refined video clip (Observation {observation_turn}):
`<video>`This video is uniformly sampled at {sample_fps:.2f} fps, contains {total_frames:.1f} frames
from {start_time:.1f} seconds to {end_time:.1f} seconds.
Continue your reasoning process inside `<think>` and `</think>`. If needed, you can keep selecting temporal segments from the original video by outputting `<grounding>` and `</grounding>` as before. Once you are ready to provide the final answer, put it inside `<answer>` and `</answer>`.

**Prompt for Forcing Answer**

After the above Action {action_turn}, here is the refined video clip (Observation {observation_turn}):
`<video>`This video is uniformly sampled at {sample_fps:.2f} fps, contains {total_frames:.1f} frames
from {start_time:.1f} seconds to {end_time:.1f} seconds.
Continue your reasoning process inside `<think>` and `</think>`. You have reached the final turn and MUST provide the final answer based on all the previous conversation and observations. Think carefully about all the information you have gathered inside `<think>` and `</think>` and give your best answer inside `<answer>` and `</answer>`. Do NOT select temporal segments from the original video as before.

*Figure 10.* **System prompt and User Prompt for Video QA.**

# G. Qualitative Analysis

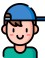

A man dressed as a clown is standing in front of a brick wall. The man has a red ball on his nose and is adorned with red decorations on his face. He is wearing a flowery ring and a purple coat. What did the man do the first time he appeared?
A. The man ran to the left side.    B. The man kept waving his hands.    C. The man wiggled his waist up and down.
D. The man spread his hands and climbed the wall.                    E. The man\'s flowery ring fell on the ground.

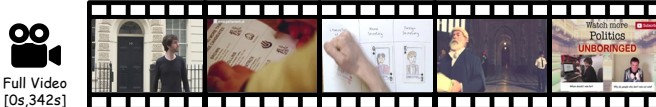

Full Video
[0s,342s]

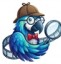

*<think>To determine what the man did the first time he appeared, I need to locate the initial scene where he is dressed as a clown. I will look for the first instance of his appearance and observe his actions. The key is to identify the first action he performs after his initial appearance.</think>*
*<grounding>{"temporal_segment": [97, 102], "sampling_strategy": "coarse"}</grounding>*

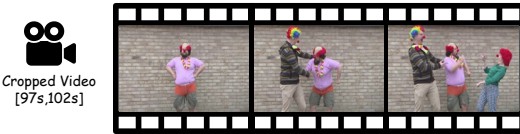

Cropped Video
[97s,102s]

*<think>The video shows a man dressed as a clown with a red ball on his nose and red decorations on his face, wearing a flowery ring and a purple coat. He is standing in front of a brick wall and begins to wiggle his waist up and down. This is the first action he performs after his initial appearance, which directly corresponds to option C.</think>*
*<answer>C</answer>*

*Figure 11.* **Single Tool Invocation Example.**

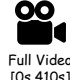

Full Video
[0s,410s]

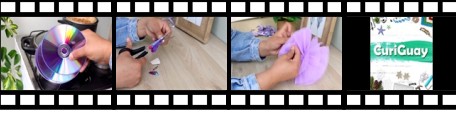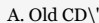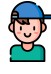

Which of the following materials is not used in this video?
A. Old CD\'s.
B. Glue or liquid silicone.
C. Satin Ribbon.
D. Wool threads.

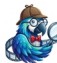

*<think>The question asks which material is not used in the video. To determine this, I need to identify all the materials shown being used. I will start by looking for the first material mentioned in the options, which is old CD\'s. I will scan the video to locate where old CD\'s are introduced and used.</think>*
*<grounding>{"temporal_segment": [0, 28], "sampling_strategy": "fine"}</grounding>*

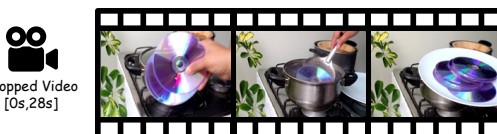

Cropped Video
[0s,28s]

*<think>The video clearly shows old CD\'s used as the primary material, as they are placed in a pot of boiling water and later dried and cut into shapes.Next, I will look for the next material mentioned in the options, which is glue or liquid silicone. I will scan the video to locate where this material is introduced and used.</think>*
*<grounding><grounding>{"temporal_segment": [68, 74], "sampling_strategy": "coarse"}</grounding>*

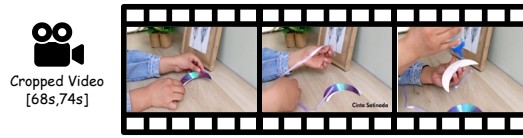

Cropped Video
[68s,74s]

*<think>The video shows the application of glue or liquid silicone to attach the satin ribbon to the cut CD shape, confirming its use in the project. Since old CD's, glue or liquid silicone, and satin ribbon are all shown being used, the only material not used in the video is wool threads.</think>*
*<answer>D</answer>*

*Figure 12.* **Multi-turn Tool Invocation with Dynamic Quota.**

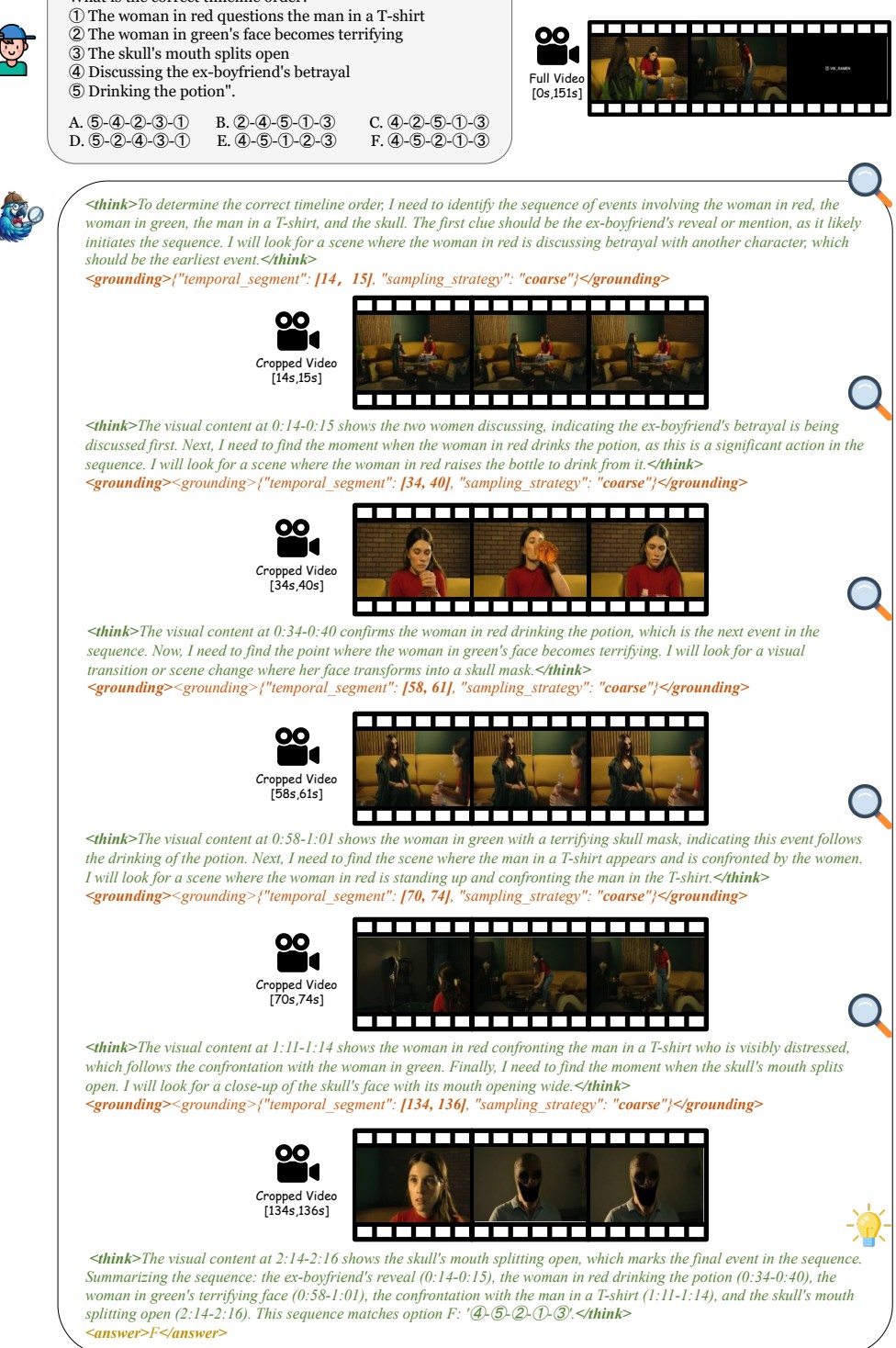

*Figure 13.* **Application of Multiple Tool Invocations in Complex Multi-hop Reasoning.**

In this section, we supplement our analysis with multiple representative qualitative results demonstrating the model's tool invocation and multi-turn interleaved reasoning capabilities. Figure 11 presents a single tool invocation example where the model precisely captures transient frames in news segments to extract crucial clues. This capability stems from Trajectory-Guided RLVR, which enables Video-o3 to exhibit strong spatial-temporal localization ability, allowing it to

answer most questions via single-round localization. Figure 12 illustrates the practical application of dynamic quota for tool invocation. The model autonomously and adaptively determines the number of visual tokens according to the length of segments to be captured, striking an effective balance between computational efficiency and detail granularity. Figure 13 demonstrates our model's robust multi-hop reasoning capability. For complex tasks such as event ordering, the model successfully gathers all necessary clues by reasoning about the question and conducting 5 consecutive rounds of precise localization.

## H. Failure Case Analysis

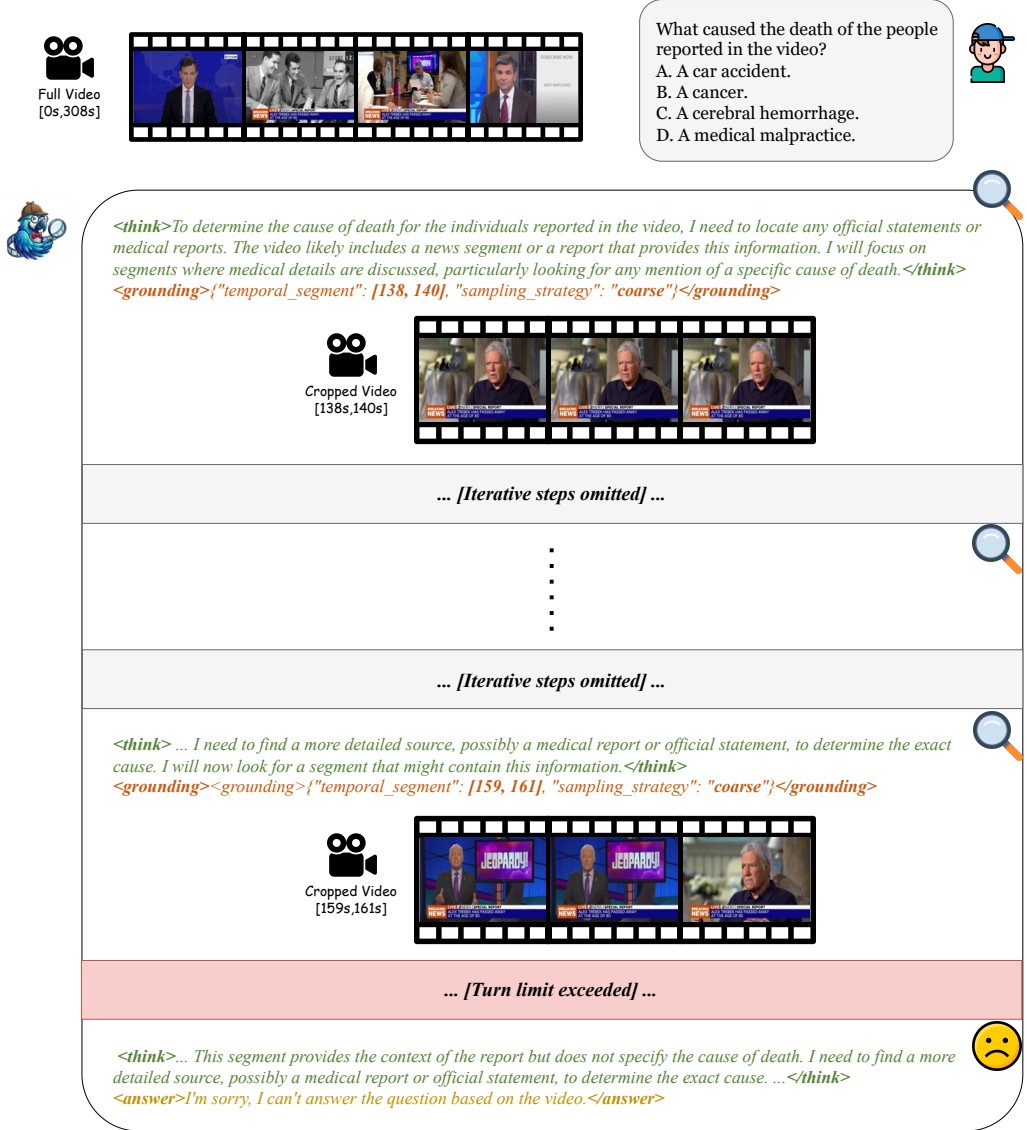

*Figure 14.* **Failure Case: Multi-turn Tool Invocation Exceeding Turn Limit**

As our approach employs a Native Multi-turn Tool Invocation paradigm where the model autonomously determines segment cropping without any hand-crafted rules, this may introduce blind exploration during clue seeking. As illustrated in Figure 14, when dealing with subtle clues in long videos, the difficulty in locating clues can cause the model to explore incorrect time intervals for too many rounds, preventing it from gathering necessary evidence within the given turn limit. Enhancing the current Native Multi-turn Tool Invocation framework with more flexible clue exploration mechanisms to support longer context processing remains as future work.

# I. Case Studies of Fake Thinking

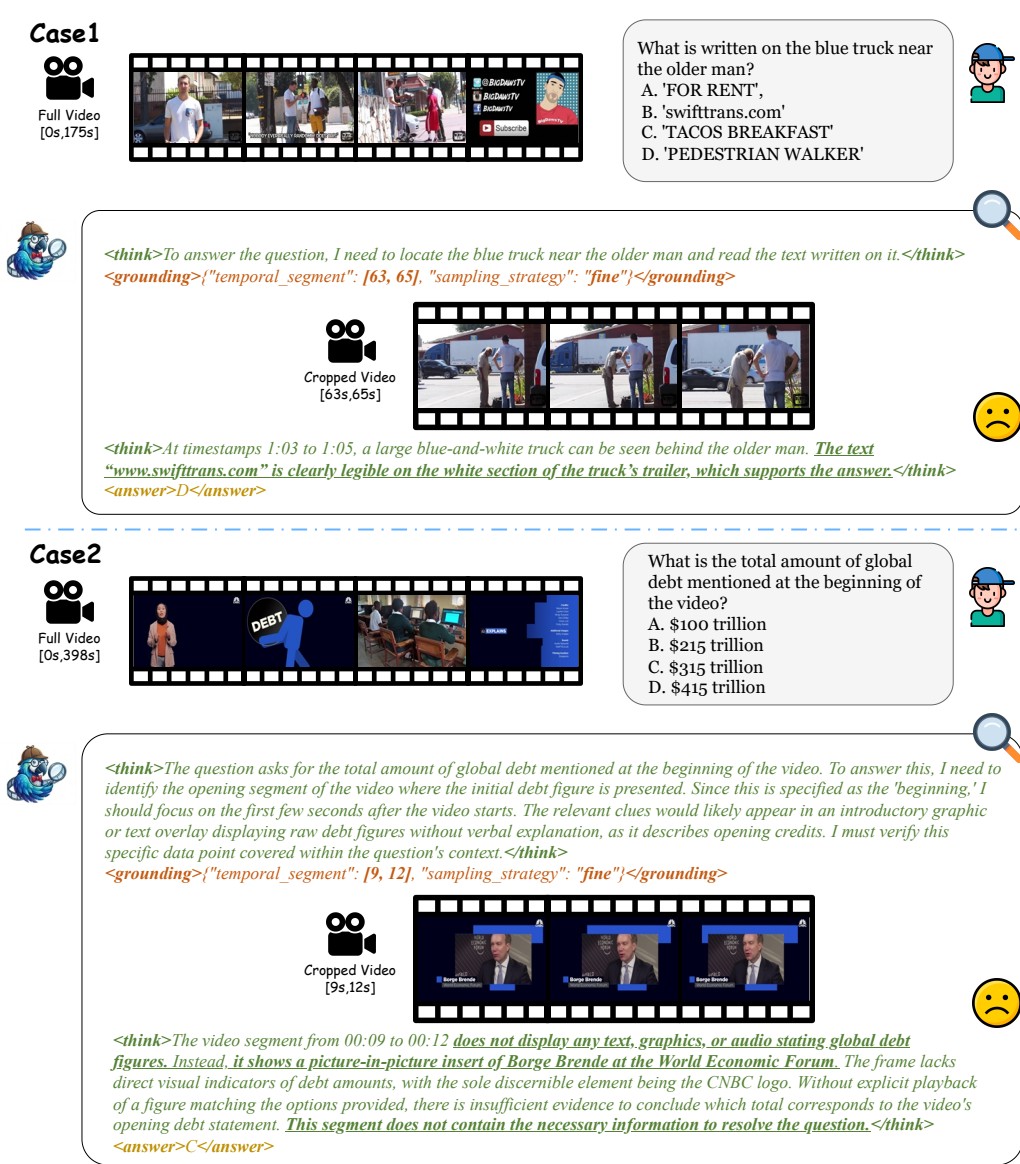

*Figure 15.* **Examples of Fake Thinking.**

In Section 3.2, we identified the "Fake Thinking" phenomenon, wherein despite successfully retrieving evidence, the model's final prediction remains inconsistent with its intermediate reasoning—that is, the model ostensibly engages in valid reasoning yet arrives at erroneous decisions (Zhu et al., 2025b). We illustrate this issue with two concrete examples. As shown in Figure 15, Case 1 (see underlined text) demonstrates that although the model successfully localizes and retrieves correct clues, it nonetheless selects an incorrect answer. In Case 2, the model initiates preliminary clue localization but fails to pinpoint the relevant evidence. Despite insufficient evidence, the model prematurely commits to an answer rather than continuing tool invocation to retrieve additional video clips. Both cases reveal that the model inadequately leverages information from the retrieved video clips and subsequent reasoning, instead disproportionately relying on the overview stage content, resulting in decisions that contradict its intermediate reasoning process.

## J. Limitations and Future Work

**Context Window and Interaction Depth.** To balance training efficiency with hardware memory constraints, we strictly limit the interaction depth to 8 turns. Although empirical results suggest this budget covers most current benchmarks, it

may fall short for extreme scenarios (Ouyang et al., 2025a; Zhu et al., 2024b), such as analyzing full-length movies or resolving queries that demand deep, multi-hop logical chains. When the context quota is depleted, the model is forced to halt inference, possibly truncating the evidence collection process. In addition, abstract puzzle-style benchmarks such as VideoReasonBench (Liu et al., 2025a) require specialized plot-reasoning supervision that is not the focus of our current training data. To address these limitations, future versions will explore dynamic context management strategies, such as pruning invalid exploration paths and converting them into lightweight error logs. This would allow the model to sustain longer reasoning trajectories without overwhelming memory resources.

**Flexibility of Tool Scalability.** While Video-o3 successfully addresses resolution bottlenecks through the VideoCrop interface, its dependence on static, predefined tool definitions limits scalability. Extending the framework to include diverse utilities (such as OCR, audio separation, or external knowledge bases) poses a substantial engineering burden, as each addition requires manually crafting specific protocols and parsers. To overcome this, we plan to transition towards a Code-as-Action paradigm. By allowing the agent to synthesize and execute Python code within a secure sandbox, we can enable flexible, on-the-fly tool usage. This approach would significantly reduce the engineering overhead associated with new tools while granting the agent superior autonomy and generalization power.

