# OpenReview forum: "Video-o3: Native Interleaved Clue Seeking for Long Video Multi-Hop Reasoning"
_ICML.cc/2026/Conference — ICML 2026 regular_

### Official Review · Reviewer_Dqpq · 2026-02-23

**Soundness:** 3
**Presentation:** 4
**Significance:** 3
**Originality:** 3
**Overall Recommendation:** 4
**Confidence:** 4

**Summary:**

The paper introduces VideoSeeker, a framework that enables end-to-end training for video understanding models with native multi-turn tool calling. To make the framework effective, the authors applied task-decoupled attention masking and verifiable trajectory-guided reward, where the former applies attention masks in SFT to shape model behaviors and the latter makes the inference more token-efficient through RL. The authors also curated a dataset Seeker-173k that enables training in this paradigm.

**Compliance With Llm Reviewing Policy:**

Affirmed.

**Ethical Review Concerns:**

This paper introduces ~15k youtube collected data (in appendix c.2.), but the authors did not disclose how data is collected (any authorizations, etc) and its intended use.

**Ethics Expertise Needed:**

["Privacy and Security (e.g., personally identifiable information)", "Legal Compliance (e.g., EU AI Act, GDPR, copyright, terms of use)"]

**Final Justification:**

The authors included additional comparison and results in the rebuttal. I maintain my already positive score and encourage the authors to included these in their revised manuscript.

**Key Questions For Authors:**

1. Task-decoupled attention masking appears to me more as an engineering compromise where authors induce certain behaviors in the attention distribution. On one hand, I wonder if authors can provide elaboration why such attention curation is needed when we should expect that the SFT objective can already steer the model to learn how to solve problems. On the other hand, I wonder if authors have tried other methods in order to steer the attention (e.g., via an objective applied to the attention instead of masking)?
2. The ultimate end goal of trajectory-guided RL should be to save tokens. While the authors' objective makes intuitive sense, I wonder how would all of these compare a much simpler baseline objective where a simple trajectory length penalty is applied as part of the reward? What is the advantage of trajectory-guided RL compared to this baseline beyond being more principled?
3. Given the self-collected datasets, I wonder if data contamination has been checked with respect to the benchmarks that the model is being evaluated on? Also, do comparisons with baseline methods have the same amount of training data?

**Limitations:**

Yes

**Strengths And Weaknesses:**

### Strengths
* A framework that enables reasoning-based multi-turn tool-use VLMs for video understanding provides a useful baseline for the community for further tool or algorithm design, which can be a huge research space.
* The dynamic visual quota for tool invocation is well-thought, which gives the model the authority to decide various parameters and at the same time is optimized toward goals (e.g., RL training).
* Both the attention masking mechanism in SFT and the reward design in RL make intuitive sense, and authors performed detailed ablations to validate their effectiveness.
* The authors collected new videos and curated tool-augmented multi-turn datasets from existing data and their collected videos. The dataset is helpful for further video understanding research in the community.

### Weaknesses
* While this framework achieved the best performance over other native multi-turn tool invocation video model baselines, it is hard to isolate whether the gains come from the framework itself or a specific data mixtures. It would be great if authors can compare the amount of source videos used & training budget compared to these baselines. Also, the bold items from VideoMMMU seem incorrect (if it indicates best performance).
* While the related work discussed both conventional reasoning and decoupled iterative reasoning paradigms for video understanding, the closet family of works, which is other baselines in open-source native multi-turn tool invocation video MLLMs, is not discussed. It would be helpful to include a detailed discussion among these baselines to understand the differences between VideoSeeker and these.
* The framework heavily depends on a curated dataset, and cannot be automatically adapted to unlabeled video datasets without extensive additional LLM annotations and verifications.

---

> ### Author Rebuttal · Authors · 2026-03-29
>
> We would like to extend our heartfelt gratitude to the reviewer for your insightful comments and valuable feedback.
>
> > **W1: Correction of Table 1 and Comparison of Training Data Volume**
>
> We will fix the formatting errors in Table 1 to accurately highlight the best-performing methods. To isolate the framework's contribution, we compare our training budget with concurrent work:
>
> |Method|Source Videos|Training Samples|VideoMME|LVBench|
> |:-|:-:|:-:|:-:|:-:|
> |Conan [CVPR26]|~31K|~91K|60.5|39.2|
> |LongVT [CVPR26]|~53K|~265K|64.3|41.3|
> |VideoSeeker (RL)|**~10K**|**~13K**|66.1|47.5|
> |VideoSeeker (SFT+RL)|~82K|~585K|66.5|47.6|
>
> VideoSeeker (RL) uses less data than Conan and LongVT, yet significantly outperforms both. While scaling data in our full SFT+RL variant yields additional gains, the substantial performance improvement in our data-efficient RL variant demonstrates that our core framework design is a primary driver of success, rather than merely the data mixture.
>
> > **W2: Discussion with Concurrent Related Work**
>
> We will expand the Related Work section to explicitly discuss native multi-turn approaches. Conan and VideoZoomer do not consider the heterogeneous patterns of tool invocation and answer reasoning. Moreover, they lack verifiable reward signals to assess the quality of tool-use trajectories. VideoSeeker advances upon these methods in two key respects: TDAM decouples clue-seeking and answer reasoning, thereby mitigating interference between heterogeneous patterns. VTGR introduces verifiable reward signals over tool invocation trajectories, enabling more precise temporal grounding and more effective early termination.
>
> > **W3: Reliance on Curated Datasets and Scalability**
>
> Although annotated trajectories facilitate rapid initial convergence, VideoSeeker can scale using Video-QA triplets without trajectory annotations through the "Free-Format Tool Invocation" task, which encourages autonomous tool strategy optimization. To demonstrate this scalability, we trained an RL model using only 15% trajectory-annotated data:
>
> |Method|VideoMME|MLVU|LVBench|
> |:-|:-:|:-:|:-:|
> |15% trajectory (RL)|63.8|70.2|43.7|
> |15% trajectory + 85% free (RL)|65.6|71.9|46.8|
> |VideoSeeker (RL)|66.1|71.9|47.5|
>
> As shown in the table, incorporating 85% unannotated trajectory data (free) recovers the majority of the performance gap, nearly matching the fully annotated baseline.
>
> > **Q1: Necessity of TDAM and Alternative Attention Objectives**
>
> We thank the reviewer for the insightful comment.
>
> **Necessity of TDAM over Naive SFT:** While standard SFT optimizes final text probabilities, it lacks the mechanism to explicitly route internal visual attention across heterogeneous expert tasks (temporal localization and answer reasoning). The disparities in representation between these two tasks often lead to contextual interference, thereby exacerbating the risk of hallucinations. TDAM addresses this by explicitly enforcing stage-specific attention patterns, achieving a structural alignment that naive SFT cannot spontaneously develop.
>
> **Why not use Attention Objectives:** Since deep attention heads focus on abstract non-local semantics, imposing rigid human-defined attention objective may lead to catastrophic forgetting. Moreover, the scalability of such an approach is limited by the prohibitive cost of generating the fine-grained, frame-and-patch-level attention labels necessary for complex multi-turn reasoning. Instead, TDAM introduces a structural constraint that effectively decouples expert tasks while preserving foundational visual representations and bypassing the need for costly attention annotations.
>
> > **Q2: Comparison with Trajectory Length Penalty and Advantage of Trajectory-Guided RL**
>
> We are glad to point out that Table 4 and Figure 8-9 in our submitted manuscript address exactly this baseline. Specifically, simple trajectory length penalty corresponds to the "w/o Hybrid Clue Score" setting in our ablation experiments. As analyzed in the paper (from left column of line 438, and lines 925-929), the primary advantage of trajectory-guided RL over this baseline is that it does not just shorten tool-calling trajectories. It significantly improves clue-searching precision, thereby directly increasing the model's answering accuracy.
>
> > **Q3: Data Contamination Check and Training Data Volume**
>
> We conducted a rigorous data contamination check prior to model evaluation. Any instances exhibiting identical source URLs or overlapping video content were strictly filtered out. Besides, we strictly ensured the use of identical training data volumes and training strategies across all ablation experiments to guarantee fair comparisons.
>
> > **Ethical Review Concerns**
>
> In strict compliance with platform policies, our dataset was sourced exclusively from public channels under the CC-BY license. The data is designated solely for academic research. We are fully committed to privacy protection and do not extract or store any PII from these videos.

---

> > ### Author Rebuttal · Reviewer_Dqpq · 2026-04-03
> >
> > Thanks for the responses and the additional comparison and results! I maintain my already positive score and encourage the authors to included these in their revised manuscript.

---

> > > ### Author Response · Authors · 2026-04-05
> > >
> > > Thank you for your constructive feedback and for confirming that our rebuttal fully resolved your concerns. We are very glad to hear that you found the additional comparisons and results helpful. As you encouraged, we will ensure that all the points discussed in our rebuttal are fully integrated into the revised manuscript.

---

### Official Review · Reviewer_ygUA · 2026-03-09

**Soundness:** 2
**Presentation:** 3
**Significance:** 3
**Originality:** 2
**Overall Recommendation:** 4
**Confidence:** 3

**Summary:**

The paper proposes VideoSeeker, a framework for long video understanding that performs native, interleaved tool invocation to iteratively seek visual evidence and reason over multiple clues. The key technical contributions are Task-Decoupled Attention Masking (TDAM) to reduce attention interference between planning and answer generation, and a Verifiable Trajectory-Guided Reward (VTGR) that balances answer correctness, localization precision, and trajectory conciseness. The authors also introduce Seeker-173K, a large-scale dataset of trajectories. The authors report strong results across multiple long video QA and reasoning benchmarks, with ablations validating the roles of TDAM and VTGR.

**Compliance With Llm Reviewing Policy:**

Affirmed.

**Final Justification:**

My initial review raised three primary concerns. First, the specific contribution of SFT in the two-stage training process was not sufficiently clear. Second, the missing results in the main table undermined the fairness and persuasiveness of the experimental comparisons. Third, while the tool-use limit was set to 8 turns, the original manuscript lacked a statistical analysis regarding the actual number of turns required and the method's scalability. Additionally, I expressed concerns regarding potential over-reliance on textual cues.

The authors have provided a comprehensive rebuttal that effectively addresses these issues. Regarding the role of SFT, the authors clarified its critical function in providing a stable cold-start for RL, thereby preventing policy degradation caused by low-quality tool calls during early training. To address the missing data in Table 1, the authors conducted additional experiments with open-source models under a unified evaluation framework, which significantly bolsters the credibility of the main results. Furthermore, the authors provided the requested statistical analysis on interaction turns and included additional results on MMR-V.

Overall, the rebuttal has clarified my main concerns. I strongly recommend that the authors incorporate these additional experiments and analyses into the final version. While I believe there is still room for further improvement in terms of expanding tool diversity and validating the approach in more complex, real-world scenarios, these do not diminish the significance of the core technical contributions. Therefore, I maintain my positive rating and have increased my confidence score.

**Key Questions For Authors:**

1. Table 1 contains many missing entries (“–”). Is this because the data is sourced from original papers? If so, it would be best to explain and summarize the settings used for each data evaluation (including sampling frequency, etc.) and other experimental configurations. However, I believe this raises questions about fairness, as different data evaluation environments affect the credibility of the paper's results.
2. Does VideoSeeker rely heavily on textual information from the questions and options to deduce the next key clue for search? How does the model handle scenarios where the clues are hidden in the connections between video frames? For example, would the model show improvements on tasks similar to those in MMR-V (ICLR 2026)[1] and VideoReasonBench[2]?
3. The article sets the maximum number of interaction rounds to 8. Is this choice primarily due to context window limitations? Did the author analyze how many rounds most tasks require to complete?
4. The core idea of this paper is to enable the model to natively support alternating between thought processes and tool calls for video inference. However, the current tool call functionality is limited to VideoCrop. Implementing video inference may require many other tools, such as those related to audio. Should more tools be added to build a better video inference framework?

[1] Zhu K, Jin Z, Yuan H, et al. MMR-V: What's Left Unsaid? A Benchmark for Multimodal Deep Reasoning in Videos[J]. arXiv preprint arXiv:2506.04141, 2025.

[2] Liu Y, Ouyang K, Wu H, et al. VideoReasonBench: Can MLLMs Perform Vision-Centric Complex Video Reasoning?[J]. arXiv preprint arXiv:2505.23359, 2025.

**Limitations:**

yes

**Strengths And Weaknesses:**

# Strength
1. The paper introduces a native interleaved tool-calling paradigm specifically designed for multi-hop reasoning in long videos.
2. The Seeker-173K trajectory dataset would be a valuable resource for the community to study native tool use in multimodal, long-context settings.
3. The paper is clearly written, with illustrations that effectively aid in understanding the proposed method.

# Weakness
1. The paper employs a two-stage training process (SFT followed by RL). However, according to Table 1, the performance gap between VideoSeeker (RL) and VideoSeeker (SFT+RL) is negligible (consistently <1%). This suggests that the contribution of the SFT to final performance may be minimal. The authors need to provide a deeper discussion on this.
2. The main results (Table 1) contain some missing entries ("-"). This likely indicates that numbers were simply cited from original papers rather than re-evaluated. This significantly weakens the persuasiveness of the results, as it prevents direct comparison on several key benchmarks. For instance, on MMVU, there are no results reported for other "Open-source Native Multi-turn Tool Invocation Video MLLMs," despite these being the most relevant works to VideoSeeker.
3. The interaction depth is limited at 8 turns, due to the context window constraint. However, the paper lacks a statistical analysis of the actual number of turns required for benchmark tasks. For complex reasoning in long movies or real-world scenarios, 8 turns may be insufficient. A discussion on the validity of this turn limitation and potential scalability solutions is missing.

---

> ### Author Rebuttal · Authors · 2026-03-30
>
> We sincerely appreciate your careful evaluation of our work and present our detailed responses as follows.
>
> > **W1: Contribution of the SFT to Final Performance**
>
> We believe the marginal performance gap between the RL and SFT+RL settings stems from the overlapping supervisory effects of SFT and trajectory-guided RL. Specifically, our trajectory-guided reward anchors critical reasoning steps by rewarding both localization accuracy and tool-use round. This effectively provides trajectory supervision that partially cover the benefits of SFT.
>
> However, SFT serves as a critical stabilizer rather than only a performance booster. As illustrated by the 'dive-and-recover' pattern in Figure 8 of our manuscript, without SFT, low-quality tool calls in early RL stages introduce noise that degrades accuracy, leading the RL policy to suppress tool invocation. SFT provides necessary cold-start that prevents the model from stagnating in shallow search behaviors and instead facilitates complex multi-hop reasoning.
>
> > **W2&Q1: Table 1 Entries and Evaluation Fairness**
>
> The values presented in Table 1 were extracted from the corresponding papers. In our revised manuscript, we will update Table 1 to explain and summarize the settings used for each model evaluation (including sampling fps and token quota). To address concerns regarding evaluation fairness, we have also re-evaluated open-source model checkpoints within our own environment (denoted as *). Due to the character limit of rebuttal, we present supplementary evaluation results for several last sota methods.
>
> |Method|Frame|Initial Visual Token|VideoMME|MLVU|LVBench|LongVideoBench|VideoMMMU|MMVU|Video-Holmes|
> |:-|:-:|:-:|:-:|:-:|:-:|:-:|:-:|:-:|:-:|
> |LOVE-R1|768|16K|64.3*|69.9*|46.7*|58.0*|50.6*|65.0*|39.5*|
> |Conan (CVPR26)|128|16K|56.9*|57.4*|37.2*|52.7*|50.2*|58.9*|44.1*|
> |VideoZoomer (ICLR26)|256|16K|64.6*|69.9*|44.0*|55.9*|51.4*|65.6*|43.8*|
> |VideoSeeker|768|16K|66.5|72.1|47.6|60.5|51.7|67.2|46.5|
>
> > **W3&Q3: Interaction Depth and Potential Scalability**
>
> Capping the tool-call interactions at 8 rounds is a trade-off intended to balance context window constraints, inference latency, and overall response accuracy. We conducted statistical analysis of the actual rounds required across 2,175 reasoning samples from long video benchmarks MLVU. The results are summarized in the table below:
>
> |Turn Count (Avg)|Turn > 2|Turn > 4|Turn > 6|
> |:-:|:-:|:-:|:-:|
> |2.31|15.4%|9.9%|1.3%|
>
> As shown in the data, only 1.3% of complex cases require more than 6 rounds. These results suggest that further increasing the round limit yields marginal performance gains.
>
> **Discussion of potential scalability solutions:** Regarding the potential for scaling to more complex long video scenarios, we envision a relevance-aware context management strategy. Specifically, once video clip is cropped via tool call, the model evaluates its relevance to the question. Any video clip deemed entirely irrelevant is immediately discarded. Should the context window exceed its budget, the system removes the video clip with the lowest relevance and replaces them with textual logs.
>
> > **Q2: Scalability of Vision-Based Complex Multi-Hop Reasoning**
>
> Thank you for your insightful question. VideoSeeker does not rely heavily on textual information. The multi-hop clue seeking in our framework is essentially a feedback loop based on joint visual and textual reasoning. Initially, the model formulates clue hypotheses by jointly analyzing the textual query and a low-resolution global video overview. The model then verifies, refines, or excludes these hypotheses through iterative high-resolution visual sampling. By dynamically sampling and comparing frames at different timestamps, the model effectively traces visual state changes and uncovers clues hidden within temporal transitions.
>
> |Method|MMR-V (Overall)|
> |:-|:-:|
> |Qwen2.5-VL-7B|32.4%|
> |VideoSeeker-7B|44.7%|
>
> We have supplemented the evaluation on MMR-V. As shown in the table above, VideoSeeker demonstrates significant and verifiable improvements, effectively bypasses the language priors by grounding its logic in actual visual observations. However, VideoSeeker performs similarly to baselines on VideoReasonBench, which tests abstract puzzles reasoning requiring specific training data that our video plot reasoning model lacks.
>
> > **Q4: VideoCrop and Tool Expansion**
>
> We agree that integrating tools like audio would further enhance performance. In this work, we prioritized VideoCrop because it addresses the most fundamental challenge in long video understanding: the precise spatial-temporal localization of visual clues, which we found to be the most critical factor for multi-hop reasoning. While adding more tools is a natural extension, it necessitates defining specialized protocols and curating extensive trajectory guided training data, which is a substantial engineering effort. We consider expanding the toolset to be a primary focus of our future work.

---

> > ### Author Rebuttal · Reviewer_ygUA · 2026-04-02
> >
> > Thank you for the detailed response. The additional experiments and explanations have effectively addressed my concerns. I encourage the authors to incorporate these new results into the revised manuscript, as they will significantly strengthen the paper.

---

> > > ### Author Response · Authors · 2026-04-05
> > >
> > > Thank you for your constructive feedback throughout the review process, and for acknowledging our rebuttal. We are very glad to hear that our additional experiments and explanations have fully resolved your initial concerns. As you suggested, we will ensure that all the new results and clarifications are incorporated into the final revised manuscript.

---

### Official Review · Reviewer_R1RR · 2026-03-11

**Soundness:** 3
**Presentation:** 3
**Significance:** 3
**Originality:** 3
**Overall Recommendation:** 4
**Confidence:** 4

**Summary:**

The paper proposes VideoSeeker, a novel long video understanding framework with multi-turn tool calling that progressively locates the salient visual clues for question answering. The "clue seeking + answer reasoning" phases are optimized end-to-end. To address the challenges in interleaved tool calling, the authors propose Task Decoupled Attention Masking (TDAM) that masks out irrelevant information for tool calling and answer generation, and Verifiable Trajectory-Guided Reward (VTGR) for balancing unconstrained autonomous
exploration and efficiency-driven trajectory regularization. To generate training data, the authors develop a data synthesis pipeline and construct the Seeker-173k dataset, with 173k tool-interaction trajectories. Experiment results show that VideoSeeker substantially outperforms state-of-the-art methods.

**Compliance With Llm Reviewing Policy:**

Affirmed.

**Final Justification:**

My concerns have been adequately addressed in the rebuttal. I will maintain my initial positive rating.

**Key Questions For Authors:**

1. The efficiency of VideoSeeker can be lower than the baselines due to multi-turn tool calling and clip analysis, especially when compared to single-turn video MLLMs. Can the authors provide quantitative comparisons on the number of visual tokens used as input and number of text tokens generated as well as the inference time for efficiency analysis?
2. The experiment is based on Qwen2.5-VL-7B-Instruct. How important is the choice of base model? Does the method generalize to smaller or weaker models?

**Limitations:**

yes

**Strengths And Weaknesses:**

Strengths:
1. The paper is well written and easy to follow.
2. The paper proposes an end-to-end framework for joint "clue seeking + answer reasoning" optimization.
3. The proposed TDAM and VTGR strategy effectively improves the model performance during SFT and RL.
4. The constructed Seeker-173k dataset can benefit the community for future research.
5. Extensive experiments demonstrate the effectiveness of the proposed method.

Weakness:
1. VideoSeeker only considers the simple VideoCrop tool. Incorporate advanced tools such as OCR and zoom-in can further improve the performance.
2. The design of Hybrid Clue Score is to improve clue localization precision and Turn Decay Factor is for agile termination. However, in the ablation study E.2., the figures only show their effect for tool call rate and average accuracy, instead of directly showing how they improve localization precision and reducing the number of turns.
3. The efficiency of VideoSeeker can be lower than the baselines due to multi-turn tool calling and clip analysis, especially when compared to single-turn video MLLMs. This should be quantify (e.g., by number of visual and text tokens, or inference time) and discuss in the paper.
4. The experiment is based on Qwen2.5-VL-7B-Instruct. How important is the choice of base model? Does the method generalize to smaller or weaker models?
5. Since the main goal of tool calling in this work is to identify the key video segments/frames, there is a long line of work on keyframe identification for long video understanding (e.g., VideoMiner ICCV25, MDP3 ICCV25, Koala CVPR24, MovieChat CVPR24 etc.) that can be included as the baselines or related works.

Minor weakeness:
1. Typo: YouTobe (Line 815)
2. The RL algorithms seem to be based on DAPO (Line 319-328) but Line 251 mentions GRPO. Although DAPO is a variant of GRPO, this needs to be clarified to avoid confusion.

---

> ### Author Rebuttal · Authors · 2026-03-29
>
> We would like to express our sincere gratitude for your review. Your suggestions will greatly help us improve our paper.
>
> > **W1: VideoCrop and Tool Expansion**
>
> We thank the reviewer for this insightful suggestion. We agree that integrating tools like OCR and zoom-in would further enhance performance. In this work, we prioritized VideoCrop because it addresses the most fundamental challenge in long video understanding: the precise temporal localization of visual clues, which we found to be the most critical factor for multi-hop reasoning. While adding more tools (e.g., OCR, audio) is a natural extension, it necessitates defining specialized protocols and curating extensive trajectory guided training data, which is a substantial engineering effort. We consider expanding the toolset to be a primary focus of our future work.
>
> > **W2: Evidence of Improve Localization Precision and Reducing Turn Number**
>
> To directly view the impact of the Hybrid Clue Score on localization precision, we conducted additional ablation on temporal grounding benchmark Charades-STA. As shown in the table below, variants incorporating the Hybrid Clue Score demonstrate consistent performance gains over those without it. Notably, the improvement is even more pronounced in the absence of temporal grounding training, which provides strong evidence for the effectiveness of Hybrid Clue Score in enhancing localization precision.
>
> |Hybrid Clue Score|Temporal Grounding Data|Charades-STA (mIoU)|
> |:-:|:-:|:-:|
> |||46.5|
> |√||53.8|
> ||√|59.6|
> |√|√|60.7|
>
> Furthermore, we performed supplementary experiments on long video benchmark MLVU to evaluate the efficacy of the Turn Decay Factor in reducing the number of turns. The results indicate that the version utilizing the Turn Decay Factor achieves the lowest average number of turns, notably suppressing ultra-long samples trapped in search loops. This validates its critical role in curbing context overhead and promoting efficient reasoning trajectories.
>
> |Stage|Turn Decay Factor|Turn Count (Avg)|Turn > 2 (%)|Turn > 4 (%)|Turn > 6 (%)|
> |:-|:-:|:-:|:-:|:-:|:-:|
> |SFT|-|2.49|24.4|16.3|2.7|
> |SFT+RL||2.57|17.6|10.4|4.8|
> |SFT+RL|√|2.31|15.4|9.9|1.3|
>
> > **W3&Q1: Token Consumption and Computational Efficiency**
>
> We appreciate this insightful comment. We present a comparison of average token consumption and inference time on the MLVU benchmark.
>
> |Model|Inference Type|Infer Time (s)|Input Visual Tokens|Response Tokens|Acc (%)|
> |:-|:-:|:-:|:-:|:-:|:-:|
> |VideoChat-R1|Single-turn|6.3|14,368|104|68.7|
> |VideoChat-R1.5|Decoupled Iterative|18.9|41,515|343|70.9|
> |LOVE-R1|Decoupled Iterative|15.2|32,108|270|67.4|
> |VideoSeeker|Native Multi-turn|10.2|18,020|235|72.1|
>
> Notably, compared to the single-turn baseline, VideoSeeker achieves a significant performance gain with acceptable increase in computational overhead. Unlike decoupled iterative reasoning methods process each turn independently, VideoSeeker operates multi-turn reasoning in a unified shared context. This naturally leverages KV-cache for eliminating redundant computation, which demonstrates that VideoSeeker can scale from single-turn to multi-turn reasoning with minimal additional cost.
>
> > **W4&Q2: Generalization Across Smaller or Weaker Models**
>
> To evaluate the generalization of our method to smaller models, we applied VideoSeeker to the Qwen3VL-4B. We compared its performance against the original Qwen3VL-4B baseline under a fixed visual token budget (~16k tokens). As shown in the table below, VideoSeeker yields consistent improvements across diverse benchmarks.
>
> |Method|VideoMME|MLVU|LVBench|LongVideoBench|VideoHolmes|
> |:-|:-:|:-:|:-:|:-:|:-:|
> |Qwen3VL-4B|66.9|72.0|46.0|61.0|37.3|
> |VideoSeeker-4B|67.3|73.2|50.0|62.5|48.7|
> |$\Delta$|+0.4|+1.2|+4.0|+1.5|+11.4|
>
> Analysis of Base Model Importance:
> Our method use the model’s temporal grounding capability as a "clue seeker" replacing traditional external keyframe selection components (e.g., CLIP-based retrieval). Consequently, the effectiveness of VideoSeeker is related to the base model's ability to perform accurate temporal localization. While our experiments demonstrate robust generalization to smaller models like Qwen3VL-4B, we acknowledge that for significantly weaker models with poor grounding performance, the tool-calling mechanism may return noisy segments, potentially limiting the performance gains. We believe a baseline level of temporal reasoning is a prerequisite for our tool-augmented paradigm.
>
> > **W5: Discussion of More Related Works**
>
> We thank the reviewer for highlighting these highly relevant works. In the revised manuscript, we will include a comprehensive discussion of these studies (e.g., VideoMiner, MDP3, Koala, and MovieChat) in the Related Work section to better contextualize our approach.
>
> > **Minor Weakness**
>
> We thank the reviewer for the careful check. We have corrected the typo to "YouTube" and unified the RL algorithm to DAPO throughout the manuscript.

---

> > ### Author Rebuttal · Reviewer_R1RR · 2026-04-01
> >
> > My concerns have been adequately addressed in the rebuttal. I encourage the authors to include the new results into the revised paper.

---

> > > ### Author Response · Authors · 2026-04-05
> > >
> > > Thank you for your positive acknowledgment of our rebuttal. We are very glad to hear that our responses and additional experiments have fully resolved your concerns. We will carefully integrate all the new results and discussions from the rebuttal into the revised paper.

---

### Official Review · Reviewer_n3pB · 2026-03-12

**Soundness:** 3
**Presentation:** 2
**Significance:** 2
**Originality:** 3
**Overall Recommendation:** 4
**Confidence:** 3

**Summary:**

This paper introduces VideoSeeker, a new framework designed to handle multi-hop reasoning in long videos. The framework uses a native interleaved tool-calling mechanism to iteratively discover visual clues, examine key segments in detail, and stop adaptively when sufficient evidence is gathered. To address two core challenges: multi-task attention dispersion and growing context length, the framework proposes a task-decoupled attention mask and a verifiable trajectory-guided reward mechanism. Additionally, this paper built a dataset containing 173K high-quality tool interaction trajectories to support both supervised learning and reinforcement learning training. Experimental results show that VideoSeeker significantly outperforms existing methods across multiple benchmarks, demonstrating its capability in multi-hop evidence search and reasoning for long videos.

**Compliance With Llm Reviewing Policy:**

Affirmed.

**Final Justification:**

My concerns have been adequately addressed in the rebuttal. I think this paper is acceptable, and I encourage the author to incorporate the relevant points from the discussion into the paper.

**Key Questions For Authors:**

1. Can you present the "free exploration" samples more clearly?
2. Since the first step feeds the entire video into the model, have you tracked the context token length or memory/FLOPs consumption compared to other models?
3. Comparisons with some of the latest models will enhance the value of this paper.

**Limitations:**

yes

**Strengths And Weaknesses:**

Strengths:
1. This paper is well-motivated; by simulating human behavior while watching videos, it attempts to enhance MLLM’s performance on long videos through active cue-seeking.
2. The paper shows some innovation, particularly in the attention mask design and the use of reward functions to enable early stopping of retrieval.

Weaknesses:
1. Some parts of the paper are hard to follow, especially the trajectory-guided multiplier component in the Verifiable Trajectory-Guided Reward (VTGR).
2. This paper emphasizes efficient context use, adaptive termination, and avoidance of redundant tool calls. But the experiments mostly report accuracy only. There is no direct evidence for reduced token consumption.
3. The experimental data are insufficient. In Table 1, there are numerous unknown indicators. I believe the authors need to complete the corresponding experiments to make the results convincing.
4. The bold formatting of the numbers in Table 1 is misleading; for example, LongVideoBench and VideoMMMU.

---

> ### Author Rebuttal · Authors · 2026-03-30
>
> We are very grateful for your constructive comments, which have been instrumental in improving our paper.
>
> > **W1: Exposition of the Trajectory-Guided Multiplier**
>
> Thank you for your constructive feedback. We will clarify our explanation of the trajectory-guided multiplier in the revised manuscript, which augments the base answer reward to incentivize optimal tool utilization. Specifically, it dynamically assigns a higher bonus as the model's predicted temporal grounding intervals and tool invocation turns align more closely with pre-annotated reference trajectories. This effectively guides the model to learn and replicate high-quality tool reasoning paths. Furthermore, data tags can be used to separately activate or deactivate the alignments of grounding intervals and turn limits, facilitating seamless transition between trajectory-guided learning and free exploration.
>
> > **W2&Q2: Token Consumption and Computational Efficiency**
>
> Since all compared methods are based on Qwen2.5-VL-7B, FLOPs are directly related to context length. We present a comparison of GPU memory, inference time (on single A100 GPU) and average token consumption on the MLVU benchmark.
>
> |Model|Inference Type|GPU Memory (G)|Infer Time (s)|Input Visual Tokens|Response Tokens|Acc (%)|
> |:-|:-:|:-:|:-:|:-:|:-:|:-:|
> |VideoChat-R1|Single-turn|18.2|6.3|14,368|104|68.7|
> |VideoChat-R1.5|Decoupled Iterative|18.5|18.9|41,515|343|70.9|
> |LOVE-R1|Decoupled Iterative|18.3|15.2|32,108|270|67.4|
> |VideoSeeker|Native Multi-turn|18.7|10.2|18,020|235|72.1|
>
> Notably, compared to the single-turn baseline, VideoSeeker achieves a significant performance gain with acceptable increase in computational overhead. Unlike decoupled iterative reasoning methods process each turn independently, VideoSeeker operates multi-turn reasoning in a unified shared context. This naturally leverages KV-cache for eliminating redundant computation, which demonstrates that VideoSeeker can scale from single-turn to multi-turn reasoning with minimal additional cost.
>
> > **W3: Table 1 Unknown Indicators**
>
> We would like to clarify that all values presented in Table 1 were directly extracted from the original publications. The missing entries (denoted by -) indicate that the corresponding paper did not report evaluation results on those benchmarks.
>
> We fully agree that providing a complete set of comparisons strengthens the persuasiveness of our work. We evaluate the open-sourced model checkpoints to fill in the missing metrics (denoted as *). Due to the character limits of rebuttal, we present the most recent works below:
>
> |Method|VideoMME|MLVU|LVBench|LongVideoBench|VideoMMMU|MMVU|Video-Holmes|
> |:-|:-:|:-:|:-:|:-:|:-:|:-:|:-:|
> |LOVE-R1|66.2|67.4|**48.2**|60.1|50.6*|65.0*|39.5*|
> |Conan (CVPR26)|60.5|63.4|39.2|56.6|50.2*|58.9*|44.6|
> |VideoZoomer (ICLR26)|65.2|69.9*|41.5|57.7|**52.2**|65.6*|43.8*|
> |VideoSeeker|**66.5**|**72.1**|47.6|**60.5**|51.7|**67.2**|**46.5**|
>
> > **W4: Table 1 Formatting**
>
> Thank you for the feedback. We have revised Table 1 to use bold text exclusively for the top-performing method on each benchmark, with this clearly noted in the caption.
>
> > **Q1: Free Exploration Samples**
>
> The core characteristic of the "free exploration" samples is that they are entirely devoid of any intermediate tool-use trajectory annotations, meaning the training supervision for the model is based solely on the correctness of the final answer. The primary motivation for introducing this subset is to compel the model to autonomously optimize its search strategies without trajectory guidance, thereby enhancing the model's generalization capabilities in unknown and complex scenarios.
>
> We designed a specific reward mechanism for the "free exploration" samples: Their Hybrid Clue Score ($S_{clue}$) is set to a constant $C_{free}$. This design differs from that of the "trajectory guided" samples (rely on the alignment between the predicted clue interval and the ground truth). The free exploration samples utilize this fixed score to encourage exploration diversity, preventing the model from being confined to predefined path.
>
> > **Q3: Comparisons with the Latest Models**
>
> We sincerely thank the reviewer for this constructive suggestion. In Table 1 of our manuscript, we have already included comparisons with the latest concurrent works, such as Conan (CVPR 2026), LongVT (CVPR 2026), and VideoZoomer (ICLR 2026). The results demonstrate that our approach outperforms these sota methods across most benchmarks.
>
> |Method|VideoMME|MLVU|LVBench|LongVideoBench|VideoHolmes|
> |:-|:-:|:-:|:-:|:-:|:-:|
> |Qwen3VL-4B|66.9|72.0|46.0|61.0|37.3|
> |VideoSeeker-4B|67.3|73.2|50.0|62.5|48.7|
> |$\Delta$|+0.4|+1.2|+4.0|+1.5|+11.4|
>
> Furthermore, we integrate our approach with Qwen3VL-4B. The performance comparison under an identical visual token budget (~16k) is presented in the table, which clearly indicate that our method continues to yield substantial performance gains even when applied to the latest base model.

---

> > ### Author Rebuttal · Reviewer_n3pB · 2026-04-03
> >
> > My concerns have been adequately addressed in the rebuttal. I think this paper is acceptable, and I encourage the author to incorporate the relevant points from the discussion into the paper. I've adjusted my score. Good luck.

---

> > > ### Author Response · Authors · 2026-04-05
> > >
> > > We sincerely thank you for your time, the constructive engagement, and for recognizing our efforts during the rebuttal phase. We are very grateful for your updated assessment and the increased score. As you suggested, we will ensure that all the points discussed during this rebuttal are carefully incorporated into the paper.

---

### Decision · Program_Chairs · 2026-04-30

**Decision:**

Accept (regular)

**Comment:**

The paper proposes VideoSeeker, a long-video understanding framework with native multi-turn tool calling. All four reviewers provided consistent positive assessments after rebuttal (Weak Accept ×4). They agree that the proposed method is novel and well-motivated. Both the attention masking mechanism in SFT and the reward design in RL are technically sound. Moreover, the constructed Seeker-173K dataset is valuable to the community.

Reviewers raised concerns regarding the clarity of SFT’s contribution in the two-stage training, the reliance on the curated dataset, and the number of tool-use turns. The authors provided a convincing rebuttal, and the reviewers generally found that most concerns were resolved.

Overall, there is a clear consensus that the paper meets the bar for acceptance. Hence, I recommend accepting this paper.